# FNOPE: Simulation-based inference on function spaces with Fourier Neural Operators

Guy Moss[1,2]*          Leah Sophie Muhle[3]          Reinhard Drews[3]

Jakob H. Macke[1,2,4,†]          Cornelius Schröder[1,2,†]*

[1]Machine Learning in Science, University of Tübingen, Tübingen, Germany
[2]Tübingen AI Center, Tübingen, Germany
[3]Department of Geosciences, University of Tübingen, Tübingen, Germany
[4]Department Empirical Inference, Max Planck Institute for Intelligent Systems, Tübingen, Germany
[†]Joint supervision

## Abstract

Simulation-based inference (SBI) is an established approach for performing Bayesian inference on scientific simulators. SBI so far works best on low-dimensional parametric models. However, it is difficult to infer function-valued parameters, which frequently occur in disciplines that model spatiotemporal processes such as the climate and earth sciences. Here, we introduce an approach for efficient posterior estimation, using a Fourier Neural Operator (FNO) architecture with a flow matching objective. We show that our approach, FNOPE, can perform inference of function-valued parameters at a fraction of the simulation budget of state of the art methods. In addition, FNOPE supports posterior evaluation at arbitrary discretizations of the domain, as well as simultaneous estimation of vector-valued parameters. We demonstrate the effectiveness of our approach on several benchmark tasks and a challenging spatial inference task from glaciology. FNOPE extends the applicability of SBI methods to new scientific domains by enabling the inference of function-valued parameters.

## 1 Introduction

Probabilistic inference of mechanistic parameters in numerical models is a ubiquitous task across many scientific and engineering disciplines. Among methods for Bayesian inference, simulation-based inference (SBI, [1–6]) has emerged as a powerful approach for performing inference without requiring explicit formulation or evaluation of the likelihood. Instead, SBI only requires a simulator model which can sample from the likelihood. By training a generative model on pairs of parameters and simulation outputs, SBI can directly estimate probability distributions such as the posterior distribution.

However, existing SBI methods are designed to infer a limited number of vector-valued parameters, which strongly limits their use for inferring spatially and/or temporally varying, function-valued parameters. In these cases, parameters are commonly inferred on fixed discretizations of the domain. Despite some recent advances leveraging generative models to infer higher-dimensional posterior

---

*{firstname.secondname}@uni-tuebingen.de
  Code available at https://github.com/mackelab/fnope

39th Conference on Neural Information Processing Systems (NeurIPS 2025).

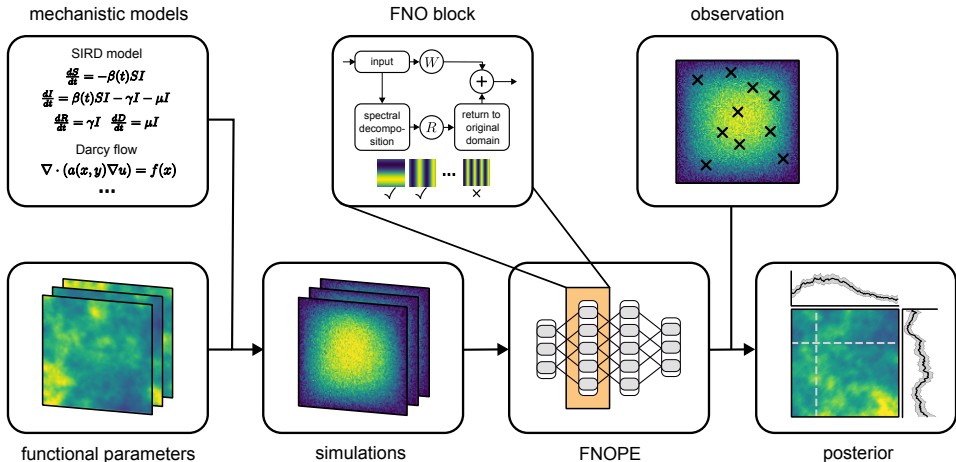

Figure 1: **Overview.** FNOPE approximates the posterior over function-valued parameters of a mechanistic model conditioned on function-valued observations. We use a FNO architecture with a flow matching objective to efficiently represent the function-valued parameters, enabling us to estimate extremely high dimensional posterior distributions at arbitrary discretizations of the domain.

distributions [7–11], the high-dimensional inference problems that arise from such approaches remain a challenge.

Furthermore, current models need to be retrained for new discretizations of the parameters or the observations. This is particularly challenging in fields like the geosciences, where observations cannot always be made at the same locations. An alternative to using fixed discretizations is to represent the functions using a fixed set of basis functions, where the inference problem becomes inferring the basis function coefficients, as used, e.g., in [12]. However, these approaches require a good selection of basis functions and suffer from a trade-off between choosing sufficiently expressive basis sets, while maintaining a tractable number of parameters to infer.

To overcome these limitations, we require methods that are capable of modeling and inferring function-valued data. Here, we propose to make use of the Fourier Neural Operator (FNO, [13]) architecture, which operates on function-valued data, for performing SBI on function-valued parameters. Neural operators [14–16] combine operations on global features of function-valued data with local (typically pointwise) operations, thus capturing both global and local structures. In particular, FNOs use Fourier features to model the global structure. For smoothly varying data, the spectral power is concentrated in the lower frequency components of the spectral decomposition. This allows for a compact representation of the global structure of the data, and hence for the inference of function-valued data on high resolution discretizations.

We present FNOPE (Fig. 1), an inference method for *function-valued parameters*: It trains **FNO**s for **P**osterior **E**stimation using a flow-matching objective [9, 17]. FNOPE is capable of solving inference problems for spatially and temporally varying parameters, and can generalize to posterior evaluations on non-uniform, previously unseen discretizations of the parameter and observation domains. Furthermore, FNOPE can estimate additional, vector-valued parameters of the simulator. We demonstrate these features on a collection of benchmark tasks, as well as a challenging real-world task from glaciology. We compare the performance of FNOPE to SBI approaches that use fixed-discretization or basis-function representation of the parameters, and show that FNOPE outperforms these methods, especially for low simulation budgets. Thus, FNOPE enables efficient inference for high dimensional inference problems that were previously challenging or even intractable.

## 2 Preliminaries

### 2.1 Simulation-based inference

SBI is designed to solve stochastic inverse problems: Given a simulator parametrized by $\theta$, a known prior distribution $p(\theta)$ and an observation $x \in \mathbb{R}^{N_x}$, the goal is to infer the posterior $p(\theta \mid x)$ for

(typically) vector-valued parameters $\theta \in \mathbb{R}^{N_\theta}$. The simulator implicitly defines the model likelihood $p(x \mid \theta)$ by allowing us to sample $x \sim p(x \mid \theta)$. We can construct a training dataset by sampling from the joint $p(\theta)p(x \mid \theta)$ to construct a dataset of simulations $S = \{(\theta_i, x_i)\}_{i=1}^K$ for a number of simulations $K$. Standard approaches in neural posterior estimation (NPE) approximate the posterior $q^\phi(\theta \mid x)$ with a normalizing flow, which is trained by minimizing the negative log-likelihood $-\mathbb{E}_{(\theta,x) \sim S} \log q^\phi(\theta \mid x)$ [1, 3]. In contrast to this, flow-matching posterior estimation (FMPE) [9] learns a conditional velocity field $v_{t,x}^\phi(\theta_t)$ to iteratively denoise samples from a base distribution (typically a Gaussian distribution) to the posterior distribution $p(\theta \mid x)$ . The velocity $v_{t,x}^\phi$ is trained via the flow matching objective

$$\mathcal{L}_{\text{FMPE}} = \mathbb{E}_{t \sim \mathcal{U}[0,1],(\theta,x) \sim S, z_t \sim p_t(z_t|\theta)} ||v_{t,x}^\phi - u_t(z_t \mid \theta)||^2, \tag{1}$$

where $p_t(z_t \mid \theta)$ are the sample-conditional flow paths for $z_t$, and $u_t(z_t \mid \theta)$ are the true velocity fields. The sample-conditional paths are chosen so that $p_t$ and $u_t$ are analytically tractable.

## 2.2 Fourier Neural Operators

We use FNOs [13] to efficiently learn the posterior distribution of function-valued parameters. FNOs are a class of neural operators using the Fourier basis as an intermediate representation of functional data to learn mappings between function spaces. We assume to have a bounded domain $D \subset \mathbb{R}^d$, on which we define function spaces $\mathcal{A}(D; \mathbb{R}^{d_a})$ and $\mathcal{B}(D; \mathbb{R}^{d_b})$. The goal of neural operators is to approximate some given operator $\mathcal{G} : \mathcal{A} \to \mathcal{B}$ by a learnable operator $\tilde{\mathcal{G}}^\phi : \mathcal{A} \to \mathcal{B}$. In practice, the function-valued data is represented as discretizations of sample functions $a_i \in \mathcal{A}, b_i \in \mathcal{B}$ on the domain $D$. A single-layer FNO, $\tilde{G}^\phi : \mathcal{A} \to \mathcal{B}$, is defined by

$$b(x) = \sigma(W^\phi a(x) + (\mathcal{K}^\phi a)(x)) \quad \forall x \in D, \tag{2}$$

where $W_\phi$ is a learnable linear operator, $\sigma$ is a (pointwise) non-linearity, and

$$(\mathcal{K}^\phi a)(x) = \mathcal{F}^{-1}(R^\phi(\mathcal{F}a))(x) \forall x \in \mathcal{D}.$$

Here, $\mathcal{F}$ and $\mathcal{F}^{-1}$ refer to the Fourier and inverse Fourier transformation, and $R^\phi$ refers to some operator acting on the Fourier modes of $a$. Typically, $R^\phi$ is a linear transformation, and therefore corresponds to a convolution in real space with $\mathcal{K}^\phi$. But typically $R^\phi$ only acts on the lower Fourier modes, discarding higher ones, and therefore gives rise to a compact representation of high-resolution data. However, as Eq. 2 includes the linear operator $W^\phi a(x)$, FNOs are still able to capture local structures.

## 3 Method

To extend the standard SBI setting to inferring function-valued parameters, we develop FNOPE by extending FMPE with FNOs as backbone (Figs. 1,2). FNOPE takes the function-valued parameters $\theta$ and observations $x$ as input, and estimates the FMPE flow-field $v^\phi$ for function-valued parameters using a combination of several FNO blocks.

We assume that $\theta$ as well as $x$ are evaluated on discretizations specified by positions $l^\theta$ and $l^x$, which means we choose it from some set of $(l^\theta, l^x) \in \mathcal{D}_\theta \times \mathcal{D}_x$. Here, the parameter positions $l^\theta$ are independent of the observation positions $l^x$ and can additionally vary between samples $i$. To adapt the parameter prior to function-valued parameters, we define a prior draw as an evaluation of an underlying measure $\mu$ (e.g., a Gaussian Process) at specific locations $l^\theta$: $\theta \sim p_{l^\theta}$. The simulator then returns observations $x$ at locations $l^x$ following the likelihood $p_{l^x}(x \mid \theta, l^\theta)$. Many such simulations create a dataset $S = \{(l_i^\theta, \theta_i, l_i^x, x_i)\}_{i=1}^K$ for a number of simulations $K$. The explicit usage of the positions $l^x$ and $l^\theta$ allows for flexible conditioning of the posterior.

### 3.1 Function-valued FMPE objective

To learn the velocity field $v^\phi$, we adapt the FMPE objective function [9] for the function-valued setting. Given a discretized observation $(x_o, l_o^x)$, and a desired parameter discretization $l^\theta$, we want to sample $\theta \sim p_{l^\theta}(\theta \mid x_o, l_o^x)$. This is done by first sampling from a base distribution $\xi_1 \sim p_{l^\theta}(\xi)$ and

then learning the velocity field $v_{l^\theta}^\phi(t, \xi_t, x_o, l_o^x)$ for $\xi_t$. In the following, we omit the arguments of $v_{l^\theta}^\phi$ for clarity. The learned velocity field allows us to iteratively denoise $\xi_1$ into a sample $\xi_0$ from the target posterior distribution. Note that the noise distribution is discretized on the same positions $l^\theta$ as the parameter $\theta$. Similarly to Eq. 1, the velocity field $v_{l^\theta}^\phi$ is optimized via the loss function

$$\mathcal{L}_1 = \mathbb{E}_{t \sim \mathcal{U}[0,1], (l^\theta, \theta, l^x, x) \sim S, \xi_t \sim p_{l^\theta}(\xi_t | \theta)} ||v_{l^\theta}^\phi - u_t(\xi_t | \theta)||^2. \tag{3}$$

Here, $p_{t,l^\theta}(\xi_t | \theta_t)$ describes a known noising process such that $\xi_1 | \theta$ is (approximately) drawn from the base distribution $p_{l^\theta}(\xi)$. Furthermore, $u_t(\xi_t | \theta)$ is the true vector field of the path defined by $p_{t,l^\theta}(\xi_t | \theta_t)$. We use the rectified flows formulation [18], such that $u_t(\xi_t | \theta_t) = (\xi_t - \theta)$.

The noise distribution, $\xi_1 \sim p_{l^\theta}(\xi)$, is commonly defined to be independent Gaussian white noise, $\xi \sim \mathcal{N}(0, I)$. Such distributions give rise to samples with a uniform power spectrum of $\xi$. As FNOs typically operate on the lower frequency modes of their inputs, independent Gaussian white noise would not be a suitable base distribution choice for our application. Instead, we sample noise from a Gaussian Process $\xi \sim \mathcal{GP}(0, k(\cdot, \cdot))$ [19, 20], where $k(\cdot, \cdot)$ is the square exponential kernel with lengthscale $l$. Using Bochner's theorem (Appendix S3.1) [21, 22], the spectral density of samples is

$$P(f) = (2\pi l^2)^{d_\theta/2} \exp\left(-2\pi^2 l^2 f^2\right),$$

where $d_\theta$ is the domain dimension of $\theta$ (and therefore of $\xi$). We choose $l$ to be dependent on the highest Fourier mode $M$ used by the FNO. This ensures that the majority of the signal power in the noise samples $\xi_1$ is conserved by the FNO block. We use the heuristic $l = \frac{2}{\pi(M/2+1)}$, which in expectation assures that $> 99\%$ of the spectral density of samples $\xi$ is in the lower $M$ frequency modes (derivation in Appendix S3.2). The covariance kernel $k$ of the Gaussian Process is scaled to have unit marginal variance and defines the FMPE noise sampling during training via

$$p_{l^\theta}(\xi_t | \theta) = \mathcal{N}((1-t)\theta, t^2 k(l^\theta, l^\theta)).$$

## 3.2 Adapting to non-uniform, unseen discretizations of the domain

To operate on non-uniform discretizations, we adopt the work of Lingsch et al. [23] and use a

Figure 2: **FNOPE architecture.** FNOPE is based on several FNO blocks (gray): A FNO block receives the discretization-dependent spectral features of the function-valued parameters and observations as an input and processes them in a linear layer before transforming it back to the original domain via the approximate inverse transformation. A pointwise linear operation on the input is added, along with embeddings of the flow time, point positions, and vector-valued parameters. We expand this setup to several parallel channels and stack layers. The vector-valued velocities are separately estimated via a MLP.

FNO with a type II non-uniform fast Fourier transform (NUDFT, [24]). The NUDFT allows us to approximate the first $M$ spectral modes of data or parameters discretized on any $N$ points in the domain. We additionally add the positions $l^\theta$ and $l^x$ to the input of the FNO blocks through multilayer perceptrons (MLPs, omitted in Fig. 2 for clarity) [25, 26].

In addition to non-uniform discretizations, FNOPE is also able to deal with distinct data and parameter discretizations at training and evaluation time. If we can query the simulator for arbitrary discretizations $l^\theta, l^x$, we can generate the training data with mixed discretizations. However, this is not the case for all simulators. To mitigate errors when evaluating the posterior at non-uniform discretizations unseen during training, we perform data augmentation during training. First, we independently mask parts of the parameters and observations by randomly removing entries of $\theta_i$ and $x_i$ and the corresponding positions $l_i^\theta$ and $l_i^x$. Second, we add small, independent Gaussian noise to the remaining positions (Appendix S3.3).

In addition to this flexible implementation, we provide and evaluate **FNOPE (fix)**, a variant of FNOPE which uses the fast Fourier transform (FFT) in the FNO blocks and can be used for applications which exclusively consider parameters and observations discretized on uniform grids. The FFT computes spectral components in $O(N \log N)$ in the number of points in the discretization $N$, compared to $O(N \cdot M^D)$ for the NUDFT, where $D$ is the domain dimension. Therefore, FNOPE scales favorably with the *parameter* dimensionality $N$ (the number of points in the discretization), but FNOPE (fix) scales favorably with the *domain* dimensionality $D$ (details in Appendix S3.4).

### 3.3 Inferring additional parameters

As most real world simulators have additional vector-valued parameters $\eta$, we extend the FNOPE architecture to also infer their posterior distribution. Vector-valued parameters are drawn from a known prior distribution $p(\eta)$ and the model likelihood becomes $p_{l_i^x}(x_i \mid \theta_i, l_i^\theta, \eta)$. The resulting inference problem is to estimate the posterior distribution $p_{l^\theta}(\theta, \eta \mid x_o, l_o^x)$. Hence, we now condition the velocity field $v_{l^\theta}^\phi$ additionally on the vector-valued parameters $\eta$ by embedding them into the channel-dimension and subsequently adding them to the input of the FNO blocks (Fig. 2).

To estimate the velocity field of the vector-valued parameters $v_\eta^\phi$ (Fig. 2), we use a multilayer perceptron (MLP) to process the spectral features of the output of the final FNO block together with the vector-valued parameters and the spectral features of the observation. This approach results in a network which targets the combined velocity $v^\phi = [v_{l^\theta}^\phi, v_\eta^\phi]$. The combined network can be trained by an extension of the loss in Eq. 3,

where the noise $p(z_t \mid \eta)$ of the vector-valued parameters is given by a normal distribution $\mathcal{N}(z_t; (1-t)\eta, t^2 \mathbf{I})$, and $u_t = [u_t(\xi_t \mid \theta), u_t(z_t \mid \eta)]$. The vector field $u_t(z_t \mid \eta)$ for the vector-valued parameters is analogously defined as $u_t(z_t \mid \eta) = (z_t - \eta)$. In practice, we separately normalize the loss for $v_{l^\theta}^\phi$ and $v_\eta^\phi$ by the number of parameters (Appendix S3.5).

## 4 Experiments

We apply FNOPE to four simulators: a Gaussian linear toy example, the SIRD model from epidemiology, the Darcy flow inverse problem and a real world application from glaciology (details in Appendix S5).

For the linear Gaussian simulator, we can analytically compute the posterior distribution, allowing us to compare the estimated posterior distributions to this ground truth using the Sliced-Wasserstein Distance (SWD) [27]. However, as is common in SBI applications, we do not have access to the ground truth posterior for the other simulators. Instead, we use a combination of two metrics to measure the quality of our posteriors: First, we report the predictive mean square error (Pred. MSE) between ground truth observations and predictive simulations from posteriors conditioned on those observations. We complement this metric with simulation-based calibration (SBC) [28] on the posterior marginal distributions. We quantify posterior calibration using the Error of Diagonal (EoD), measuring the average distance of the calibration curve of the estimated posterior from a perfectly calibrated posterior. Good performance on both of these metrics is not a sufficient condition to indicate a correctly estimated posterior, but healthy posteriors typically achieve good performance on these metrics. All evaluation metrics are averaged over three runs and we report mean $\pm$ standard error. We provide more details on all evaluation metrics in Appendix S2. We provide an overview of training and sampling times, as well as network sizes and computational resources used for all tasks, in Appendix S1.

### 4.1 Baseline methods

We compare FNOPE to three baseline methods: NPE (with normalizing flows) [3] and FMPE [9] on the coefficients of the spectral basis functions of the parameters (NPE/FMPE (spectral) respectively, details in Appendix S4). We also compare to FMPE with a fixed parameter discretization (FMPE (raw)).

For all baseline methods, we use the *sbi toolbox* [29]. For the SIRD simulator we compare to Simformer [30], a transformer-based amortized inference approach that is also capable of flexible

discretization of function-valued parameters. The other baselines cannot be applied in their basic version to this task because they do not support non-uniform discretizations of both parameters and observations.

## 4.2 Linear Gaussian

We first show the ability of FNOPE to approximate the true posterior of a linear Gaussian, as commonly done in SBI benchmarks [31]. To illustrate FNOPE's ability to infer a large number of parameters, we increase the dimensionality to 1000. We also replace the independent Gaussian prior in this task with a Gaussian process to model smoothly-varying function-valued parameters.

FNOPE clearly outperforms all benchmark methods on this problem (Fig. 3). With a training dataset of $10^2$ simulations the SWD is close to zero for both FNOPE and FNOPE (fix). In contrast, both NPE and FMPE based on spectral features need as many as $10^5$ simulations to achieve similar performance. Furthermore, this example shows that the data augmentation applied in training FNOPE, results in a small difference between FNOPE and FNOPE (fix), which is an effect of the introduced positional noise. FNOPE learns a posterior under a slightly broader likelihood than what is defined by the model and

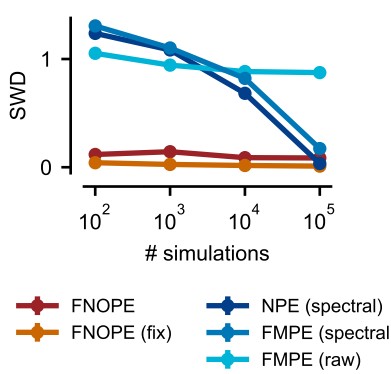

Figure 3: **Linear Gaussian simulator.** Sliced-Wasserstein distance (SWD) to ground truth posterior.

for very constrained posteriors the posterior quality is slightly poorer. However, we will see that for more challenging tasks, this is an acceptable trade off, as we gain flexibility on evaluation points.

We perform ablation experiments (Appendix S7.1), and observe that the performance of FNOPE is dependent on using sufficiently many Fourier modes in the FNO blocks (Fig. S1a,b). However, other hyperparameters show less influence on the performance (Fig. S1c-e).

## 4.3 SIRD: Inference on unseen, non-uniform discretizations

Next, we consider the Susceptible-Infected-Recovered-Deceased (SIRD) model [32] to demonstrate the ability of FNOPE to solve inference problems on non-uniform discretizations of the parameters and observations that were not seen in the training data. In addition, we also show its ability to simultaneously infer vector-valued parameters. The model has three parameters: recovery rate, death rate, and contact rate [33, 34]. We use the same setup as in Gloeckler et al. [30], where we assume that the contact rate varies over time, but recovery and death rates are constant in time. We sample training simulations on a dense uniform grid for both parameters and observations. For evaluation we sample 100 observations, each discretized on a different set of 40 randomly sampled time points in $[0, 50]$, using contact rates defined on a distinct set of 40 randomly sampled times.

FNOPE, as well as Simformer, can reliably infer the posterior distribution (Fig. 4a) and the observations lie close to the mean of the posterior predictive (Fig. 4b). Both methods are comparable in terms of MSE of posterior predictive samples to the observations, as well as producing well-calibrated posteriors (Fig. 4c). When we use only 20 timepoints to condition on, the performance of FNOPE slightly decreases (Fig. S2). This highlights the necessity of the FNO block to have enough observation points to perform a reliable (approx.) Fourier transformation. We also observe that FNOPE performs robustly across the base distribution lengthscale (Fig. S3). This experiment shows that FNOPE is on par with the state of the art on this low dimensional problem: It successfully infers function-valued parameters together with vector-valued parameters and can be conditioned on arbitrary discretizations of the observations. However, FNOPE can also be applied to very high dimensional problems, as shown in the following experiment.

## 4.4 Darcy flow: Scalable inference in high dimensions

The Darcy flow is defined by a second order elliptic PDE and has been used to model many processes including the deformation of linearly elastic materials, or the electric potential in conductive materials.

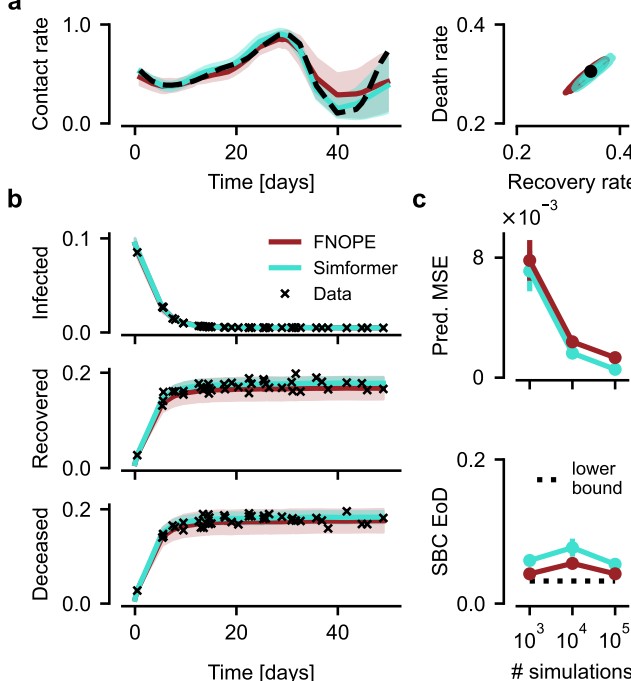

Figure 4: **SIRD model.** **(a)** Posterior conditioned on 40 time points. *left:* Posterior (mean ± std.) of the time-varying parameter and ground truth parameters (dashed). *right:* Two dimensional posterior of vector-valued parameters and ground truth parameters (dot). **(b)** Posterior predictive (mean ± std.) of infected, recovered and deceased populations with observations marked. **(c)** *upper:* MSE of posterior predictive samples to observations. *lower:* Simulation-based calibration error of diagonal (SBC EoD). 'Lower bound' refers to the SBC EoD for uniformly sampled posterior ranks (details in Appendix S2).

In the geosciences, the Darcy equation is used to describe the distribution of groundwater as a function of the spatially variable hydraulic permeability, which can be inferred from point observations in wells [35]. The Darcy flow is a common benchmark model for FNO applications, especially in the context of training PDE emulator models [13, 36, 37].

We consider the steady-state of the two dimensional Darcy flow equation on a unit square:

$$-\nabla \cdot (a(x)\nabla u(x)) = 1 \qquad\qquad x \in (0,1)^2$$
$$u(x) = 0 \qquad\qquad x \in \partial(0,1)^2,$$

where $a(x) \geq 0$ is the permeability we want to infer and $u(x)$ is the hydraulic potential. We adapt the implementation from [38], which provides a GPU-optimized solver. We use a log-normal prior distribution for the permeability, similar to Lim et al. [37]: $b = \log(a) \sim N(0, (-\Delta + \tau I)^{-2})$, where $\Delta$ is the Laplacian operator and $\tau = 9$. We sample the prior on a $129 \times 129$ grid which results in $\approx 16$k parameters. For FNOPE, we use the first 32 Fourier modes in both spatial dimensions and for spectral NPE/FMPE we used the first 16 modes, resulting in $2 \cdot 16^2 = 512$ parameter dimensions. For all methods, we infer the log-permeability and evaluate in the original space (as in [37]).

Samples from the posterior inferred with FNOPE closely resemble the ground truth (Fig. 5a and Fig. S6). Both FNOPE and FNOPE (fix) correctly capture the fine-structure of the posterior samples and reproduce parameters at much higher fidelity than all baseline methods. While the spectral methods learn oversmoothed posteriors that do not capture local structures, the posterior samples from FMPE (raw) are much noisier and only capture the rough global structure. The posterior means show a similar trend (Fig. S7), and the standard deviations of the baseline methods are higher compared to FNOPE (Fig. S8).

The MSEs between posterior predictive samples and ground truth observations of FNOPE and FNOPE (fix) are consistently better compared to the spectral baseline methods, especially at lower simulation budgets, and are in the same range as FMPE (raw) (Fig. 5b). While all methods are reasonably well-calibrated (Fig. 5c), the visual appearance is vastly different. We additionally measure the posterior quality in terms of posterior log-probability (normalized by the number of pixels) of the associated ground truth parameter $\theta$ [31]. FNOPE has a much higher log probability (Fig. 5d) compared to FMPE (raw). FNOPE (fix) also achieves strong performance for a sufficient number of simulations. As the spectral methods do not model the parameters directly, we cannot calculate the log-probabilities they assign to the ground truth parameters. Overall, FNOPE is the only

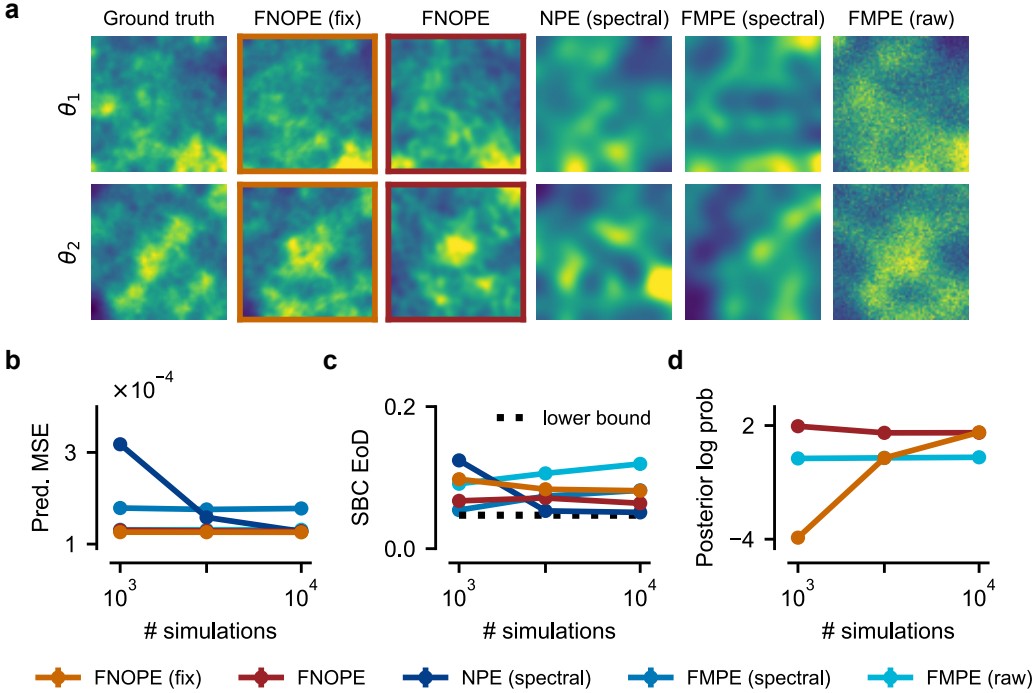

Figure 5: **Darcy flow. (a)** Ground truth parameter and posterior samples for a simulation budget of $10^4$ training samples (more posterior samples in Fig. S6). **(b)** MSE of posterior predictives to the ground truth observation (FNOPE, FNOPE (fix) and FMPE (raw) visually overlap). **(c)** Simulation-based calibration Error of Diagonal (EoD) for 50 dimensions. **(d)** Posterior log-probability of ground truth samples normalized by the number of dimensions (higher is better).

method that consistently performs well on all presented metrics. In additional ablation experiments (Fig. S4), we show that FNOPE attains strong performance across different hyperparameter choices, and that our lengthscale heuristic (Sec. 3.1) is an appropriate choice for this task.

## 4.5 Mass balance rates of Antarctic ice shelves: Real world application

Finally, we turn to a real-world task from glaciology: Inference of snow accumulation and basal melt rates of Antarctic ice shelves from radar internal reflection horizons (IRHs) [39–41]. Snow continuously accumulates on top of the ice shelf. Over time, it is transported to larger depths where the former surfaces are further deformed by ice flow and form internal layers of constant age, which are measured by radar (Fig. 6a,b). The inference of accumulation and melt rates is a challenging SBI task, where the model is misspecified, as it cannot accout for all real-world effects. We use an isochronal advection scheme forward model as described in [39]. In this work, the authors consider simulations on a grid of 500 points along a one-dimensional spatial domain and directly infer 50 parameters on a fixed downsampling of this domain using NPE. We refer to this approach as NPE (raw) and compare it to FNOPE and the baseline methods.

First, we evaluate all methods on a test set of simulations (Fig. 6c). As with the previous tasks, the performance of FNOPE at $10^3$ simulations is comparable to the performance of the other methods at $10^5$ simulations in terms of predictive MSE, and is only marginally worse at $10^2$ simulations. All methods show a reasonable calibration in terms of SBC EoD at all simulation budgets (Fig. 6d). We then test the performance of all methods on real data (as in [39]). Posterior predictive samples from FNOPE match the observation very well (Fig.6b), and while FNOPE still performs better at low simulation budgets than the other methods, the relative improvement compared to the baselines is smaller than the one observed on synthetic data (Fig. 6e). We note that this modeling problem was explicitly set up by Moss et al. [39] so that NPE (raw) can infer the posterior using a feasible number of simulations. FNOPE achieves the same performance with two orders of magnitude fewer

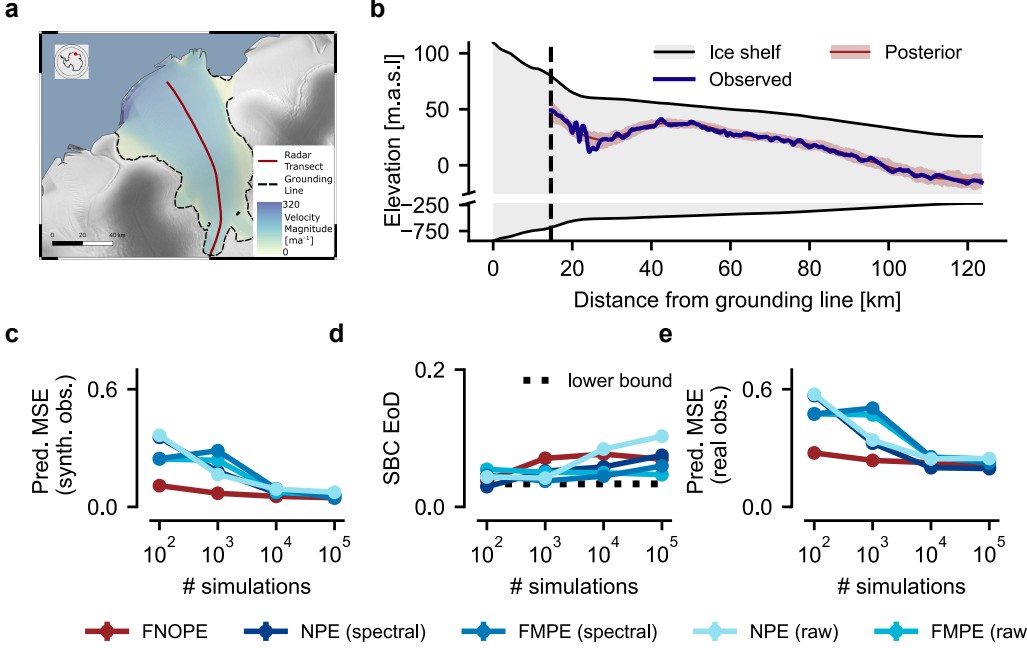

Figure 6: **Mass balance rates of ice shelves.** **(a)** Measurement transect of the radar data in Ekström Ice Shelf, Antarctica (adapted from [39], published under Creative Commons CC BY license). **(b)** Posterior predictive results obtained with FNOPE (trained on $10^5$ simulations), compared to radar-based observation on one layer in the ice shelf. **(c)-(e)** Performance measures on test simulations and the real observation, where NPE (raw) refers to the method used in [39].

simulations. Additionally, FNOPE is able to infer the full parameter dimensionality (500) instead of downsampling to 50 dimensions (Fig. S5).

## 5  Discussion

We present FNOPE, a simulation-based inference method using Fourier Neural Operators to efficiently infer function-valued parameters. On a variety of task, we showed that FNOPE can infer posteriors for function-valued data at very small simulation budgets compared to baseline methods, especially for high-dimensional problems. In addition, by building upon existing work for FNOs on non-uniform discretizations of the domain, FNOPE can generate samples from posteriors and observations defined on any discretizations of the domain, even if these discretizations were never seen during training.

**Related work**    Scaling SBI methods to high-dimensional parameter spaces has been the focus of many works which make use of state of the art generative modeling techniques such as generative adversarial networks [7], diffusion models [8, 10, 11] and flow matching [9]. In particular, recent works [30, 42] use a transformer architecture to tokenize function-valued parameters, allowing for complete flexibility in estimating conditional distributions. However, as these methods explicitly model each point for discretized function-valued parameters, they are limited in terms of scalability. Our FNO-based approach allows us to compactly represent the parameters, significantly lowering the computational costs as the number of parameters grows. Finally, for applications where only the one-dimensional marginal distributions of the Bayesian posterior are needed, it is possible to scale SBI methods to higher dimensions, as the correlation between the parameters does not need to be captured [43]. However, in most scientific applications the correlation structure is an essential object of interest and marginal distributions are only of limited use.

Estimating function-valued parameters using FNOs [44–46] and other neural operators [47–49] has been explored in previous work. A majority of these approaches consider deterministic inversion, or estimating a single value for the parameters as opposed to targeting the Bayesian posterior. While

probabilistic generative models such as invertible Fourier Neural Operators (iFNOs, Long et al. [50]) estimate a conditional distribution, they do not explicitly target the Bayesian posterior. We compare the performance of iFNO on the Darcy task, and while it has comparable performance in terms of predictive MSE, the uncertainty estimates are not well-calibrated and the conditional distribution collapses to a tiny parameter region (Appendix S8).

The closest neighbours to our work are Lingsch et al. [51], who use a FNO architecture with an FMPE objective to learn vector-valued parameters and, additionally, learn an emulator producing function-valued observations. The crucial difference to our approach is that their simulators are deterministic and the inferred parameters are not function-valued. Recently, Lim et al. [37] developed an approach for score-based modeling in function spaces using FNOs. This enables high-dimensional posterior inference, but their approach is limited to uniform grids and does not allow for flexible conditioning.

Another related approach is diffusion posterior sampling [52–54], which seeks to learn a high-dimensional prior distribution from samples using score-based models. The learned priors can then be used to generate samples from the posterior using analytically tractable model likelihoods [55, 56]. Other works extended such approaches for intractable likelihoods [57, 58]. Similarly to diffusion posterior sampling, we only require prior samples. However, instead of learning the prior distribution, we learn the posterior distribution directly.

**Limitations**    The FNO-backbone used by FNOPE  inherently makes assumptions about the structure of the parameters.  These assumptions enable computationally efficient inference, but result in some limitations. First, the FNO assumes limited high-frequency information in the parameter and observation domains. Therefore, they are ill-suited to infer parameters with high power in higher frequencies—for example, parameters with discontinuities. This could potentially be addressed by neural operators using other transforms, such as wavelet transforms [59]. Furthermore, to (accurately) compute the FFT or NUDFT of the observations, we require sufficiently many points in their discretization. Therefore, unlike other flexible methods [30, 42], our approach cannot perform inference using extremely sparse observations.  Still, the SIRD experiment shows that even for 20 points we get reasonable estimations (Fig. S2).  Finally, the computational complexity of our approaches still scales exponentially with the domain dimension, which could be challenging in high-dimensional domains.

**Conclusion**    We presented FNOPE, a simulation-based inference approach for inferring function-valued data. FNOPE can be applied to non-uniform, unseen discretizations of the domain, can scale to large parameter dimensions, and can be trained using comparatively small simulations budgets. As we show in various experiments, FNOPE can therefore tackle spatiotemporal inference problems that were previously challenging or even intractable for simulation-based inference.

## Acknowledgments and Disclosure of Funding

This work was funded by the German Research Foundation (DFG) under Germanys Excellence Strategy  EXC number 2064/1  390727645 and SFB 1233 Robust Vision (276693517) and DFG (DR 822/3-1) and the Heinrich-Böll-Stiftung. Data collection was supported by Alfred Wegener Institute through logistic grants AWI_ANT_18. The authors acknowledge support by the state of Baden-Württemberg through bwHPC. GM is a member of the International Max Planck Research School for Intelligent Systems (IMPRS-IS). We thank Manuel Gloeckler for providing the training data and Simformer results for the SIRD experiment. We thank Daniel Gedon and Julius Vetter, and all members of Mackelab for discussions and feedback on the manuscript.

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

# Supplementary Material

## S1 Software and Computational Resources

For all baseline SBI methods, we use the `sbi` toolbox [29], for the Simformer baseline we use the publicly available code from Gloeckler et al. [30] . We use an optimized solver to solve the Darcy Flow PDE [38].

We use various compute resources for the different experiments. For each experiment, we run the training and evaluation for each method, for each simulation budget, and for each of the three random seeds separately. The summary of network sizes and compute times for all tasks are provided in Tab. S1,S2,S3.

For the Linear Gaussian and SIRD experiments, we perform our experiments on Nvidia RTX 2080ti GPU nodes. Both simulators have negligible wall-clock costs on these GPU nodes. The Darcy flow experiment required GPUs with higher VRAM to accommodate the large ($\approx 16k$ dimensional) parameters and observations. We performed these experiments on Nvidia A100 GPUs. For the Antarctic Ice experiment, we perform training and evaluation on CPU, namely Intel Xeon Gold 16 cores, 2.9GHz. We perform this experiment on CPU as the main computation cost is in running the simulations for the predictive MSE check, and the implementation of the model is not accelerated by use of GPUs. The simulation costs are described in Moss et al. [39].

Table S1: **Network sizes and training times for Gaussian Linear (GL) and Darcy Flow (Darcy) tasks.** We report the mean over 3 runs with a training budget of 10k simulations. Baselines estimating the "raw" (untransformed) parameters are denoted (R) and those estimating Fourier coefficients are denoted (S). Linear Gaussian task was run on a 2080ti GPU, Darcy flow on an A100 GPU.

| Task | Metric | FNOPE | FNOPE (fix) | NPE (S) | FMPE (S) | FMPE (R) |
|------|--------|-------|-------------|---------|----------|----------|
| GL | # params. | 110K | 109K | 567K | 86.3K | 299K |
| GL | train (Tot.) [m] | 3.12 | 1.98 | 6.77 | 2.46 | 5.14 |
| GL | train (/epoch) [s] | 0.99 | 0.87 | 2.17 | 0.24 | 0.31 |
| GL | sample (/sample) [ms] | 2.16 | 1.06 | 0.05 | 0.24 | 0.20 |
| Darcy | # params. | 11.6M | 11.6M | 3.54M | 898K | 9.18M |
| Darcy | train (Tot.) [m] | 72.2 | 41.8 | 20.5 | 10.9 | 12.2 |
| Darcy | train (/epoch) [s] | 20.3 | 26.2 | 2.82 | 0.66 | 0.73 |
| Darcy | sample (/sample) [ms] | 280 | 35.2 | 0.21 | 1.95 | 2.70 |

Table S2: **Network sizes and training times for SIRD.** We report the mean over 3 runs with a training budget of 10k simulations. Both methods were trained and evaluated on a 2080ti GPU.

| Task | Metric | FNOPE | Simformer |
|------|--------|-------|-----------|
| SIRD | # params. | 117K | 286K |
| SIRD | train (Tot.) [m] | 6.95 | 47.17 |
| SIRD | train (/epoch) [s] | 1.19 | 9.43 |
| SIRD | sample (/sample) [ms] | 0.22 | 0.22 |

Table S3: **Network sizes and training times for Antarctic Ice (Ice) task.** We report the mean over 3 runs with a training budget of 10k simulations. Baselines estimating the "raw" (untransformed) parameters are denoted (R) and those estimating Fourier coefficients are denoted (S). All methods were trained and evaluated on CPU.

| Task | Metric | FNOPE | NPE (S) | FMPE (S) | NPE (R) | FMPE (R) |
|------|--------|-------|---------|----------|---------|----------|
| Ice | # params. | 25.3K | 236K | 92.3K | 361K | 96.2K |
| Ice | train (Tot.) [m] | 47.4 | 19.7 | 31.6 | 6.80 | 33.5 |
| Ice | train (/epoch) [s] | 9.82 | 4.21 | 1.92 | 5.64 | 2.02 |
| Ice | sample (/sample) [ms] | 22.8 | 1.50 | 16.6 | 1.53 | 23.9 |

## S2 Evaluation details

We here describe more details about our evaluation procedures. We evaluate on a heldout test set $\{(\theta_j^o, l_j^\theta, \eta_j^o, x_j^o, l_j^x)\}_{j=1}^{J_\text{test}}$, where $J_\text{test}$ is the number of test simulations. Given an approximate posterior distribution $q_{l^\theta}^\phi(\theta, \eta \mid x, l^x)$ and a test observation $(x_j^o, l_j^x)$, we draw $K_\text{post}$ posterior samples $(\theta_{kj}, \eta_{kj}) \sim q_{l^\theta}^\phi(\theta, \eta \mid x_j^o, l_j^x)$. In the case where no vector-valued parameters $\eta$ are present, they can be omitted. Similarly, for methods which do not explicitly use the positions $l^\theta, l^x$ (e.g. FNOPE (fix)), the positions can be omitted, as we do not apply these methods to tasks where we consider arbitrary discretizations.

We report the average and standard error over all $J_\text{test}$ test simulations.

### S2.1 Sliced Wasserstein Distance

Following Bonneel et al. [27], we define the (empirical) sliced Wasserstein(-2) distance (SWD) between $N$ samples from two probability distributions $p$ and $q$ as

$$\text{SWD}(p, q) = \mathbb{E}_{u \sim U(\mathbb{S}^{D-1})} \left[ \left( \frac{1}{K} \sum_{i=1}^{K} ||x_u^{(k)} - y_u^{(k)}||_2^2 \right)^{1/2} \right], \tag{4}$$

where $x_k \sim p(x), y_k \sim q(y)$ are samples from the two distributions, $u$ are uniformly randomly sampled vectors on the unit sphere $\mathbb{S}^{D-1}$, and $x_u^{(k)}, y_u^{(k)}$ are the 1-dimensional $i$-th order statistics of the projections $u^\top x_k, u^\top y_k$ respectively. We calculate the SWD with 50 random projections $u$ and $K = 1000$ posterior samples.

### S2.2 Simulation-based Calibration Error of Diagonal

Simulation-based calibration (SBC) [28] is a standard measure of the calibration of approximate posterior distributions (in terms of over- or underconfidence). We obtain ranks $r_{ij}$ for each sample $(\theta_j^o, x_j^o)$ in the test set using SBC with the 1-dimensional marginal distributions used as the reducing functions. That is, for each of the dimensions $i$ of $\theta$, the rank $r_{ij}$ is an integer in $(1, K_\text{post} + 1)$. This results in $J_\text{test}$ ranks. The cumulative distribution function of ranks is therefore

$$\text{CDF}_i(\alpha) = \frac{1}{J_\text{test}} \sum_j \mathbb{I}[r_{ij}/K_\text{post} < \alpha].$$

The SBC Error of Diagonal (SBC EoD) is then the mean absolute distance between this cumulative distribution and the cumulative distribution function of a uniform distribution,

$$\text{SBC EoD}(i) = \int_0^1 |\text{CDF}_i(\alpha) - \alpha| d\alpha.$$

In contrast to the SBC area under the curve (SBC AUC), the EoD will detect poor calibrations for posteriors that are overconfident at low confidence levels $\alpha$, and underconfident at high $\alpha$ (or vice-versa).Finally, we report the average SBC EoD across the dimensions of $\theta$,

$$\text{SBC EoD} = \frac{1}{N_\theta} \text{SBC EoD}(i).$$

For SIRD, since the posterior dimensionalities are low, we compute an average SBC EoD over all one-dimensional marginals of the posterior. For the Darcy Flow and Antarctic Ice tasks, we select a subset of 50 marginal distribution for computing the SBC EoD, regularly spread across the domain.

### S2.3 Predictive MSE

To calculate the MSE for posterior predictive samples, we run for each posterior sample $(\theta_{kj}, \eta_{kj})$, and each true observation $x_j^o$, the simulator $x_{kj} \sim p_{l_j^x}(x \mid \theta_{kj}, l^\theta, \eta_{kj})$. We then compute the average mean square error of the simulation $x_{kj}$ to the corresponding observations $x_j^o$,

$$\text{MSE} = \frac{1}{J_\text{test} K_\text{post}} \sum_{k=1, j=1}^{J_\text{test}, K_\text{post}} \frac{1}{|l_j^x|} ||x_{kj} - x_j^o||_{L^2}^2,$$

where $|l_j^x|$ is the number of points in the discretization $l_j^x$ and therefore the dimensionality of $x_j^o$. We use this metric since the simulators considered in this work correspond to (unknown) unimodal likelihood functions—a correctly estimated posterior will produce simulations clustered around the true observation. We opt for this metric to quantify predictive performance due its clear interpretability. However, for multimodal likelihood functions, this metric can be replaced with a scoring rule.

## S2.4 Posterior Log Probability

For the Darcy Flow task, we additionally report the posterior log-probability of the true parameters [31], normalized by the number of pixels:

$$\text{log-probability per pixel} = \frac{1}{|l_j^\theta|} \log q_{l_j^\theta}^\phi(\theta_j^o \mid x_j^o, l_j^x).$$

For the spectral methods NPE/FMPE (spectral), we cannot directly compute the posterior-log-probabilities, as we can only compute the posterior-log-probabilities of the first $M$ modes of the spectral decomposition of the ground truth parameters $\theta_j^o$. However, by discarding the information of the higher modes, we remove the information which these baseline methods cannot capture, thus biasing the resulting log-probabilities in favor of these baselines.

# S3 FNOPE details

## S3.1 Bochner's Theorem

We state Bochner's theorem following Williams and Rasmussen [22]. A complex-valued function $k$ on $\mathbb{R}^d$ is the covariance function of a weakly stationary mean continuous complex-valued random process on $\mathbb{R}^D$ if and only if it can be represented as

$$k(\tau) = \int_{\mathbb{R}^D} \exp^{2\pi i f \cdot \tau} d\mu(s) \tag{5}$$

for some positive finite measure $\mu$. Crucially, in the less general but relevant case that $\mu$ admits a density $P(f)$, the integral is a Fourier transform between the kernel $k(\tau)$ and the spectral density $P(f)$. We apply this result to relate the lengthscale of the square exponential kernel to the spectral density of its Fourier decomposition in Sec. 3.1.

## S3.2 Kernel lengthscale heuristic

The spectral density of samples from a Gaussian Process with a square exponential kernel of lengthscale $l$ is stated in Sec. 3.1 as

$$P(f) = (2\pi l^2)^{d_\theta/2} \exp(-2\pi^2 l^2 f^2).$$

This is also a Gaussian density, and trivially we see that the full spectral power is

$$\bar{P} = \int_{\mathbb{R}^{d_\theta}} P(f) df = 1$$

We consider discretizations normalized to $[0, 1]$ in each dimension, and so the power contained in the first $M$ spectral modes, $\bar{P}_M$, corresponds to the above integral within the domain $||f||_\infty \leq M/2$, i.e. where all the components of $f$ are within $[-M/2, M/2]$. Therefore $\bar{P}_M$ simplifies to the product of the Gaussian integrals

$$\bar{P}_M = \left( \int_{-M/2}^{M/2} (2\pi l^2)^{1/2} \exp(-2\pi^2 l^2 f_i) df_i \right)^{d_\theta}$$

$$= \left( \text{erf} \left[ \frac{\pi l M}{\sqrt{2}} \right] \right)^{d_\theta}$$

$$= \left( \text{erf} \left[ \frac{M\sqrt{2}}{M/2 + 1} \right] \right)^{d_\theta},$$

where erf is the Gauss error function, and in the last line we substituted our heuristic $l = \frac{2}{\pi(M/2+1)}$. This value saturates the error function and produces values very close to 1. For example, for the Darcy Flow example ($d_\theta = 2$), we set $M = 32$. The resulting spectral power in the first 32 modes is $\bar{P}_{32} \approx 0.9997$. While individual samples from the Gaussian process can result in discrete Fourier transforms where this spectral property is not fulfilled, it is clear that the majority of the spectral power will be contained in the first $M$ modes for all samples.

## S3.3   Flexible discretization

We provide further detail of the data augmentation scheme introduced in Sec. 3.2. First, we describe why this is necessary despite the use of the non-uniform fast fourier transform (NUDFT). The NUDFT is applied as a matrix multiplication., $\Theta = V(l^\theta)\theta$, where $\Theta$ is a vector containing the first $M$ spectral components of $\theta$, and $V$ is the discretization-dependent transformation matrix. The inverse NUDFT is similarly implemented as a matrix multiplication $\theta = \bar{V}^\top(l^\theta)\Theta$, where $\bar{V}^\top$ is the conjugate transpose of $V$. This approach enables the computational efficiency of the NUDFT, as the exact inverse matrix does not need to be computed at runtime. However, $\bar{V}^\top$ is only an approximate inverse of $V$, with the approximation error increasing for increasing non-uniformity of the discretization.

Consider the common case where the simulation dataset $S$ (Sec. 3) provides parameters $\theta_i$ and observations $x_i$ always discretized on the same, uniform simulation domain. Without data augmentation, we always apply the NUDFT and its inverse without approximation error. However, if we wish to condition a posterior on $x^o$ measured at some non-uniform discretization $l^x$, then the NUDFT and its inverse will produce some error, which was unseen during training. This could lead to unpredictable, out of distribution errors at evaluation time. By explicitly passing the positions $l^\theta$ and $l^x$ to the network, as well as augmenting them to ensure the network is not always applied on uniform discretizations during training, we give FNOPE capacity to learn to counteract these approximation errors.

**Masking**   We define a uniform distribution over the binary mask vectors with a fixed number of nonzero entries, $N_{\mathrm{ds}}$. Suppose we are given a simulation $(\theta, l^\theta, x, l^x)$, where $\theta, l^\theta$ consist of $N_\theta$ points, and $x, l^x$ consist of $N_x$ points. We construct two random binary masks, $\mathbf{M}^\theta \in \{0,1\}^{N_\theta}, \mathbf{M}^x \in \{0,1\}^{N_x}$, each with exactly $N_{\mathrm{ds}}$ nonzero entries. We then remove the corresponding elements of $\theta, l^\theta$ where $\mathbf{M}_\theta$ is zero, and similarly remove the corresponding elements of $x, l^x$ where $\mathbf{M}_x$ is zero. For a minibatch of simulations, we independently sample the masks $\mathbf{M}_i^\theta, \mathbf{M}_i^x$ for each simulation. If $N_{\mathrm{ds}} > N_\theta$ or $N_{\mathrm{ds}} > N_x$, we leave the corresponding value and position vector unchanged. The value of $N_{\mathrm{ds}}$ used in our work is reported for each experiment in Appendix S6.

**Positional noise**   We additionally add small, independent gaussian noise $\epsilon_i \sim \mathcal{N}(0, \sigma^2)$ to each point in $l^\theta$ and $l^x$ in the unmasked positions. This reduces generalization error for simulation datasets where the discretization of $\theta$ and $x$ is fixed. The value of $\sigma^2$ can be set according to the spacing of the discretization. In our experiments, we always normalize the simulation domain to $[0, 1]$ in all dimensions, and set $\sigma = 10^{-3}$.

## S3.4   Fixed discretization

For applications with uniform grids, we provide FNOPE (fix). Here, we use the FFT instead of the NUDFT in the FNO blocks to transform the data from physical to spectral space and back. In addition, we do not mask any parameters during training and do not add any positional noise. We expect this method to have improved performance on uniform grids as we do not introduce additional noise through the data augmentation process described above.

Another potential advantage of FNOPE (fix) is its computational efficiency. Consider the case of a parameter discretized uniformly in a $D$-dimensional domain, with $L$ points per dimension, leading to a total of $L^D$ points. The computational cost of computing the spectral decomposition of the parameters using the FFT is $O(L^D \log L^D) = O(DL^D \log L)$. Assuming the maximum number of modes in each dimension modeled by the FNO is $M$, this leads to $M^D$ total modes to compute. Therefore, the computational cost of the frequency-limited NUDFT is $O(M^D L^D)$. For one-dimensional domains, the NUDFT may well be faster to compute than the FFT. However, for

higher dimensions, the NUDFT scales exponentially with both $M$ and $L$. This discrepancy increases with both the dimensionality of the domain, and the number of modes $M$ modeled by the FNO blocks.

### S3.5    Additional Parameters

The näive extension of the FMPE objective as stated in Sec. 3.3 is to minimize the $L^2$ loss $||v^\phi - u_t||^2$, where $v^\phi = [v_{l^\theta}^\phi, v_\eta^\phi]$ and $u_t = [u_t(\xi_t|\theta), u_t(z_t|\eta)]$. However, the scale of the loss for the continuous parameters, $||v_{l^\theta}^\phi - u_t(\xi_t|\theta)||^2$ varies with the number of points in the discretization $l^\theta$, which we denote $N_\theta$. To ensure that the loss is balanced for the function- and vector-valued parameters, we in practice add their $L^2$ losses, normalized by their respective vector dimensionalities. That is, we minimize

$$\mathbb{E}\left[\frac{1}{N_\theta}||v_{l^\theta}^\phi - u_t(\xi_t|\theta)||^2 + \frac{1}{N_\eta}||v_\eta^\phi - u_t(z_t|\eta)||^2\right],$$

where $N_\eta$ is the fixed dimensionality of $\eta$. The expectation is over the same random variables as for the statement of the loss $\mathcal{L}_2$ in Sec. 3.3.

## S4    Baseline Methods

### S4.1    Spectral preprocessing

For spectral NPE/FMPE we first apply a Fourier transformation to the parameters, take the first $M$ Fourier modes and expand these $M$ complex values to a real vector of dimension $2M - 1$ representing the real and imaginary parts (the imaginary part of the first component is always 0, hence it is discarded). For two dimensional data, the same preprocessing results in $2M^2 - 1$ parameters. After inferring the posterior of the parameters in Fourier space and sampling from it, we apply the inverse Fourier transform to get samples in the spatial domain.

For the one dimensional problems (Linear Gaussian and Ice Shelf) we first pad the data by replicating the first/last value and perform a real FFT with `torch.fft.rfft`. We then use the first $M$ fourier components and expand these complex numbers to a real tensor of dimension $2M$, which we use as input to NPE/FMPE. For the Linear Gaussian and Ice Shelf we use $M = 50, 10$, respectively. We then revert this process for samples from the posterior with `torch.fft.irfft` with the corresponding settings.

For the two dimensional Darcy flow, we use the two dimensional FFT implemented in *pytorch* on the padded data (in mode replicate). We then center the frequencies before cropping to the first $M$ Fourier components in both dimensions, and expanding it to a real tensor of dimension $2M^2$. For posterior samples we again revert this process with the corresponding settings.

### S4.2    NPE (spectral)

For NPE (spectral) we infer the posterior over the coefficients of the first $M$ Fourier modes following the spectral preprocessing described above. We use NPE [3] with normalizing flows. We do not apply spectral preprocessing to the observations but pass raw observations through an embedding net as it is common practice in NPE.

### S4.3    NPE (raw)

For the mass balance experiment (Sec. 4.5), we also compare to the approach of Moss et al. [39], which we refer to as NPE (raw). This approach infers the mass balance parameters on a fixed discretization of 50 gridpoints. The authors use a Neural Spline Flow with 5 transformations, two residual blocks of 50 hidden units each, ReLU nonlinearities and 10 bins. The embedding net used to embed the 441-dimensional observation is a CNN with two convolutional layers of kernel size 5, with ReLU activations and max pooling of kernel size 2. The convolutional layers are followed by two linear layers with 50 hidden units and output dimension 50. The same settings are used in the 500-dimensional experiment (Appendix S7.4). Training is performed with a batch size of 200 and an Adam optimizer with learning rate of 0.0005.

### S4.4 FMPE (spectral)

As with NPE (spectral), in this approach we apply spectral preprocessing to the parameters, and infer the coefficients of the top $M$ Fourier modes, but process the observations directly using an embedding net. We use MLPs to estimate the flows, as in Wildberger et al. [9], as implemented in the `sbi` toolbox [29]. This implementation also uses the rectified flow [18] objective for FMPE, and we use independent Gaussian noise as the noise distribution. The flow networks are conditioned on time by concatenating the time to the inputs.

### S4.5 FMPE (raw)

In this approach, we infer the parameters directly on a fixed discretization of the domain. We again use embedding nets to encode the observations, and MLPs to learn the flows. As with FMPE (spectral), we use a rectified flow objective, and independent Gaussian noise as the noise distribution, as in the `sbi` toolbox. The flow networks are conditioned on time by concatenating the time to the inputs.

### S4.6 Simformer

For the SIRD experiment, we apply Simformer with the same settings as in Gloeckler et al. [30]. That is, we use a transformer model with a token dimension of 50, 8 layers, and 4 heads. The widening factor is 3, and the training was performed with a batch size of 1000 and an Adam optimizer. We train Simformer to learn all conditionals, and so uniformly draw between the posterior, joint, and likelihood masks, as well as two random masks drawn from Bernoulli distributions with $p = 0.3$ and $p = 0.7$ respectively. For both SIRD experiments, where we evaluate on 20 and 40 time points respective, we use the same Simformer model which is trained using 20 randomly sampled time points.

## S5 Simulators

### S5.1 Linear Gaussian model

The Gaussian Simulator is inspired by Lueckmann et al. [31], but instead of a 10 dimensional Gaussian distribution with independent dimensions we expanded the problem to 1000 dimensions and use a Gaussian Process prior (see below). Draws from the simulator are still drawn independently per dimension as $x \sim \mathcal{N}(\theta, \sigma^2 \boldsymbol{I})$, where $\sigma^2 = 0.1$ (as in [31]).

**Prior**    The prior is defined as a Gaussian process $\mathcal{GP}$ on $[0, 1]$ with an equidistant discretization with 1000 timepoints. A draw from the prior is therefore defined as $\theta \sim \mathcal{GP}(0, k(\cdot, \cdot))$, where $k$ is the squared exponential kernel, $k(t, t') = \exp(-\frac{(t-t')^2}{2l^2})$. We set the lengthscale $l = 0.05$ and variance to 1. Only in the ablation experiments (Fig. S1a,b) we changed the lenghtscale $l$ to 0.005.

**Evaluation parameters**    The results for Fig. 3 is based on 100 observations and 1000 posterior samples for each observation.

### S5.2 SIRD model

Similar to Gloeckler et al. [30] we extend the SIRD (Susceptible, Infected, Recovered, Deceased) model to have a time-dependent contact rate. Compared to the classical SIR framework the model additionally incorporates a deceased (D) population. Similar models were explored by Chen et al. [33], Schmidt et al. [34]. This addition is important for modeling diseases with significant mortality rates. The SIRD model, including a time-dependent contact rate $\beta(t)$, is defined by the following set

of differential equations:

$$\frac{dS}{dt} = -\beta(t)SI,$$

$$\frac{dI}{dt} = \beta(t)SI - \gamma I - \mu I,$$

$$\frac{dR}{dt} = \gamma I,$$

$$\frac{dD}{dt} = \mu I.$$

Here, S, I, R and D are the susceptible, infected, recovered, and deceased population, $\beta(t)$ is the time dependent contact rate, and $\gamma$ and $\mu$ are the recovery and mortality rates among the infected population. We simulate on a dense uniform grid of 100 time points for parameters and observations. The simulations are additionally contaminated by an observation noise model, which is described by a log-normal distribution with mean $S(t)$ and standard deviation $\sigma = 0.05$.

**Prior**  We impose the same prior as in Gloeckler et al. [30]: the global variables $\gamma$ and $\mu$ are drawn from a uniform distribution, $\gamma, \mu \sim \text{Unif}(0, 0.5)$. For the time-dependent contact rate we define a Gaussian process prior which is further transformed by a sigmoid function to ensure that $\beta(t) \in [0, 1]$ for all $t$. For the Gaussian process we use a RBF kernel $k$ defined as $k(t, t') = \exp(-\frac{\|t-t'\|^2}{2 \cdot 7^2})$.

**Evaluation parameters**  MSE as well as SBC EoD is based on 100 observations with 1000 posterior samples each (Fig. 4c and Fig. S2c). To calculate the posterior predictive, we sample the initial condition $I(0)$ from the Simformer prediction for both methods, as opposed to the prior defined above, to match the setting of Gloeckler et al. [30].

## S5.3  Darcy Flow

Details for the Darcy model are already given in the main text. We additionally scale the log-permebility $a$ by the scale factor of 1000 before taking the exponential - this results in permeabilities which produce sufficiently variable solutions using the Darcy flow simulator, and the permeabilities on the same scale as reported in Lim et al. [37]. The simulation output is additionally corrupted by an independent Gaussian observational noise per pixel $\zeta_i \sim \mathcal{N}(0, \sigma_i^2)$. We set $\sigma_i = \mathbb{E}[u_i^2/30]$, where the expectation is per simulation batch, resulting in a signal to noise ratio (SNR) of 30.

**Evaluation Parameters**  All metrics are calculated over a test set of 10 observations (Fig. 5 b-d). For MSE and SBC EoD, 100 posterior samples were used for each observation and for each method. Finally, for SBC EoD, we use a subset of 50 pixels of the full $129 \times 129$ as the marginals used for the reducing functions. The same pixels were used across all methods and all observations.

## S5.4  Mass balance rates of Antarctic Ice Shelves

We use the same simulator as described in Moss et al. [39]. This model takes in spatially varying surface accumulation rates $\dot{a}(l)$, which are related to the basal melt rates $\dot{b}(l)$ through the total balance condition $\dot{m}(l) = \dot{a}(l) - \dot{b}(l)$. $\dot{m}(l)$ is known and fixed across all simulations.

The layer prediction model consists of a set of isochronal layer with prescribed thicknesses $\{h_1(l), h_2(l), \ldots, h_K(l)\}$ such that $\sum_{k=1}^K h_k(l) = h(l)$, where $h(l)$ is the known and fixed total ice shelf thickness. At each time step, the thickness of the layers are simultaneously updated through an advection equation,

$$\frac{\partial h_k}{\partial t} = -\nabla \cdot (h_k u),$$

where $u(l)$ is the known velocity profile of the ice shelf which is fixed across simulations. Additional layers are added at the top of the ice shelf and removed from the bottom according to $\dot{a}(l), \dot{b}(l)$ accordingly. The noise model approximates the observation noise of the radar measurement given an assumed density profile of the ice shelf. At the final timestep $T$, the layer that most closely matches

the ground truth observation $x^o$ according to the $L^2$-norm is selected as the simulator output. Full details are described in Moss et al. [39].

**Prior** The prior is defined over the accumulation rate parameter $\dot{a}(l)$, and is motivated by physical observations at Ekström ice shelf. Prior samples are drawn using

$$\dot{a}(l) = \sigma_{\mathrm{sc}}\dot{\alpha} + \mu_{\mathrm{off}}, \tag{6}$$

where $\mu_{\mathrm{off}} \sim \mathcal{N}(0.5, 0.25^2)$, $\sigma_{\mathrm{sc}} \sim \mathcal{U}([0.1, 0.3])$, and $\dot{\alpha}$ is drawn from a Gaussian Process with a unit-variance, zero-mean Matérn-$\nu$ kernel of lengthscale 2500 and $\nu = 2.5$.

**Evaluation Parameters** For SBC EoD, as well as the predictive MSE on synthetic test simulations, we use 100 test observations and sample 10 posterior samples for each observations and for each method. The real test data consists of one field observation (shown in Fig. 6a-b), and the posterior predictive was estimated using 1000 posterior samples.

# S6 Experimental details

## S6.1 Linear Gaussian

For all the baseline methods, we train the networks using an Adam optimizer with a learning rate of 0.0001, and a batch size of 200. For NPE/FMPE (spectral), we use 50 modes, leading to 100 parameters to learn, and a pad width of 20 for the spectral preprocessing (Appendix S4.1). For NPE (spectral) the density estimator is a Neural Spline Flow (NSF) with 2 residual blocks with 50 hidden dimensions each, 5 transforms, with RELU activations. For FMPE (spectral), we use an MLP with 5 linear layers with 64 hidden dimensions to estimate the flow, with ELU activations. In both cases, we embed the 1000 dimensional observation into a 40-dimensional vector using an MLP with 2 layers and 50 hidden units and RELU activations. For FMPE (raw), we use an MLP with 5 layers and 64 hidden features, with ELU activations.

For FNOPE and FNOPE (fix) we use 50 Fourier modes for the FNO blocks. We use 5 FNO blocks with 16 channels, while the context is embedded into 8 channels. We train for a maximum of 500 epochs with an early patience of 50. We used a training batch size of 512 and a learning rate of 0.001. For FNOPE, we use 4 channels each for the positional and time embeddings and the target gridsize $N_{\mathrm{ds}} = 256$ (for FNOPE (fix), no positional embedding is included). All nonlinearities are GELUs.

## S6.2 SIRD

For FNOPE we use 32 Fourier modes for the FNO blocks. We use 5 FNO blocks with 16 channels, while the context is embedded into 8 channels. We train for a maximum of 1000 epochs with an early patience of 50. We use a training batch size of 200 and a learning rate of 0.001. The discretization positions and flow times are embedded into 4 channel dimensions each, and the target gridsize $N_{\mathrm{ds}} = 40$. This experiment additionally included vector-valued parameters in $\mathbb{R}^2$. These are embedded into a 16-dimensional vector using a 1-layer MLP with a hidden dimension of 64. The flow for the vector-valued parameters is estimated using an 1-layer MLP with a hidden dimension of 64. The spectral decomposition of the output of the FNO blocks, as well as the spectral decomposition of the observation, are embedded into a 32-dimensional vector and concatenated to the input of the MLP. All nonlinearities were GELUs.

The training hyperparameters for simformer are described in S4.6.

## S6.3 Darcy Flow

For all the baseline methods, we train the network using an Adam optimizer with a learning rate of 0.0001, and a batch size of 200. For NPE/FMPE (spectral), we use 16 modes, leading to $2 \times 16^2 = 512$ parameters to learn, and a pad width of 20 in each dimension for the spectral preprocessing (Appendix S4.1). For NPE (spectral) the density estimator is a NSF with 2 residual blocks with 50 hidden dimensions each, 5 transforms, with RELU activations. For FMPE (spectral), we use an MLP with 8 layers with 256 hidden dimensions to estimate the flow, with ELU activations. For FMPE (raw), we use an MLP with 8 layers and 256 hidden features, with ELU activations. All

baseline methods embed the observation with a CNN embedding net into a 100-dimensional vector using 4 convolutional layers with kernel size 5 followed by max pooling of kernel size 2, followed by a 4-layer MLP with 100 hidden units, with RELU nonlinearities throughout.

For FNOPE and FNOPE (fix) we use 32 Fourier modes for the FNO blocks. The network is made of 5 FNO blocks with 32 channels, while the context is embedded into 32 channels. We train for a maximum of 300 epochs with an early patience of 50. We use a training batch size of 200 and a learning rate of 0.0005. For FNOPE, we set the target gridsize $N_{ds} = 2048$. The architecture includes 8 channels for positional embedding, and 8 channels for time embedding. All nonlinearities are GELUs.

### S6.4   Mass balance rates of Antarctic Ice Shelves

For all the baseline methods, we train the network using an Adam optimizer with a learning rate of 0.0001, and a batch size of 200. For NPE/FMPE (spectral), we use 10 modes, leading to 20 parameters to learn, and a pad width of 20 for the spectral preprocessing (Appendix S4.1). For NPE (spectral) the density estimator is a NSF with 2 residual blocks with 50 hidden dimensions each, 5 transforms, with RELU activations. For FMPE (spectral), we use an MLP with 5 linear layers with 64 hidden dimensions to estimate the flow, with ELU activations. For FMPE (raw), we use an MLP with 5 layers and 64 hidden features, with ELU activations. For all baseline methods, we used the same embedding as Moss et al. [39], which was a CNN embedding the 441-dimensional observation into a 50-dimensional vector using 2 convolutional layers with kernel size 5 followed by max pooling of kernel size 2, followed by a 2-layer MLP with 50 hidden units, with RELU nonlinearities throughout. The configuration of NPE (raw) is described in Appendix S4.3.

For FNOPE  we use 10 Fourier modes for the FNO blocks. The network is made of 5 FNO blocks with 16 channels, while the context is embedded into 8 channels. We train for a maximum of 1000 epochs with an early patience of 50. We use a training batch size of 200 and a learning rate of 0.001. We do not include the data augmentation procedure for this experiment, as the discretizations of both observations and parameters was fixed to the setting of [39]. We still include positional embedding due to the parameter and observations being discretized differently to one another: the architecture included 4 channels for positional embedding, and 4 channels for time embedding. All nonlinearities are GELUs.

For the experiments on 500 gridpoints (Fig. S5) we use the same hyperparameters.

## S7   Ablation experiments

### S7.1   Linear Gaussian

To investigate the influence of different hyperparameters on the performance of FNOPE, we ran several ablation experiments. First, we studied the performance of FNOPE in the deliberately insufficient setting where the prior distribution over the parameter contains higher-frequency modes than what is modeled by the FNO blocks. To this end, we changed the lengthscale of the prior distribution for the Linear Gaussian task (Sec. 4.2) by a factor of 10 to 0.005, resulting in higher frequency components for the parameter $\theta$. We then trained FNOPE with a varying number of spectral modes in the FNO blocks. With a lower number of modes, the performance of FNOPE degrades, and the posterior samples clearly miss the high frequency components present in the ground truth observations (Fig. S1a,b). Second, we varied the number of unmasked points in the FNOPE training procedure (Sec. 3.1). While the SWD decreases with the number of unmasked points, it saturates at 512 unmasked points, which is approximately half of the observed data (Fig. S1c). We then consider changing the lengthscale heuristic used to define the base distribution for FNOPE (Sec. 3.1), by defining $l = L_0/M$ for the number of modes $M$ and some lengthscale scaling factor $L_0$. In the extreme case $L_0 = 0$, this corresponds to the base distribution sampling uncorrelated white noise (WN). The lengthscale heuristic used in our work corresponds approximately to $L_0 = 4/\pi \approx 1.27$. In contrast to the other hyperparameter ablation experiments, varying the lengthscale scaling factor does not seem to impact the performance of FNOPE considerably for this task. FNOPE performs well over a wide range of lengthscale scaling factors (Fig. S1d). Finally, we consider a version of FNOPE which keeps the masking scheme as described in Sec. 3.1, but without adding positional noise to the remaining positions, which effectively only sees positions on the simulation grid during training

(FNOPE (no jitter)). On this task, we evaluate on an equispaced grid, and we see that FNOPE (no jitter) performs similarly to FNOPE (Fig. S1e). Therefore, the performance of FNOPE does not degrade from the inclusion of positional noise in this task.

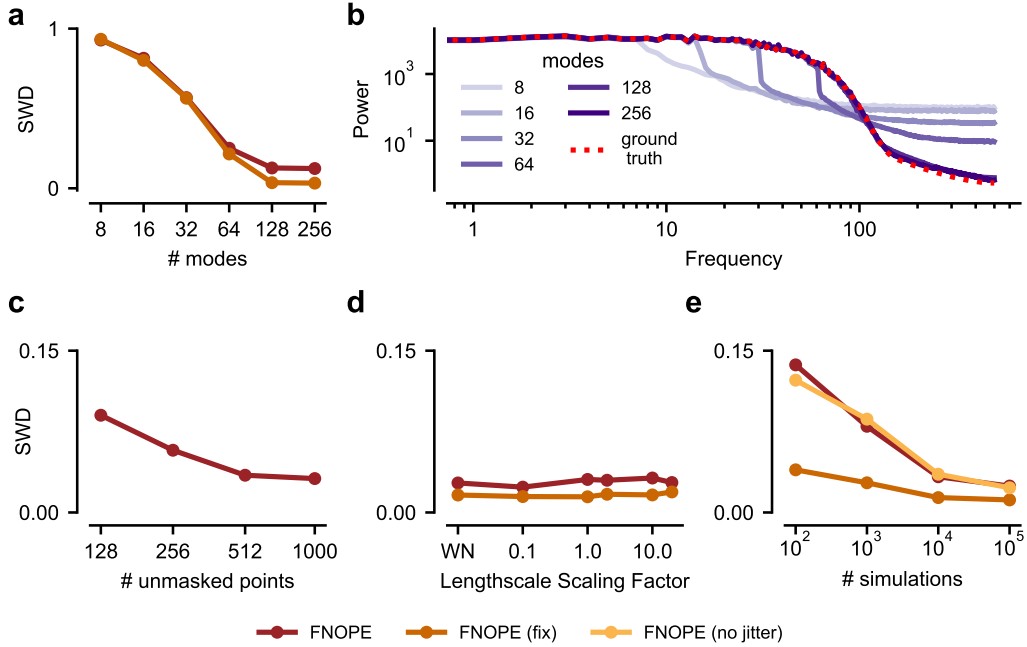

Figure S1: **Linear Gaussian ablation experiments.** **(a)** Performance for $10^4$ training samples in terms of SWD for the Linear Gaussian experiments with varying number of Fourier modes in the FNO block. Note that we changed the lengthscale of the simulator prior by a factor of 10 to 0.005 (Appendix S5. **(b)** Power analysis of FNOPE (fix) samples and ground truth $x$ for different number of used modes in the FNO block. **(c)** Performance in terms of SWD for different numbers of masked points in FNOPE. **(d)** Influence of the lengthscale of the noise process on the SWD. "WN" corresponds to white noise (an uncorrelated Gaussian distribution). **(e)** Same as Fig. 3 with a version of FNOPE in which we omit the adding of positional noise (FNOPE (no jitter)).

## S7.2   SIRD

First, we compare the performance of FNOPE to Simformer on the SIRD task when using observations at 20 (instead of 40) randomly sampled times (Fig. S2). The performance of FNOPE slightly degrades, while Simformer still performs robustly. Second, we adapt the lengthscale heuristic used to define the base distribution for FNOPE (Sec. 3.1), by defining $l = L_0/M$ for the number of modes $M$ and some lengthscale scaling factor $L_0$. The lengthscale heuristic used in our work corresponds approximately to $L_0 = 4/\pi \approx 1.27$. We observe that FNOPE performs robustly for a wide range of lengthscale scaling factors.

## S7.3   Darcy Flow

In the Darcy task, we ablate the number of modes used in the FNO block for FNOPE. We see that while the MSE of posterior predictive simulation and posterior calibration as measured by the SBC EoD of marginal distributions is not strongly affected by the number of modes used, the posterior log-probability per pixel increases with the increased model capacity (Fig. S4a). This is expected, as more modes allow FNOPE to estimate a more constrained posterior distribution, leading to higher posterior densities. Second, we adapt the lengthscale heuristic used to define the base distribution for FNOPE (Sec. 3.1), by defining $l = L_0/M$ for the number of modes $M$ and some lengthscale scaling factor $L_0$. The lengthscale heuristic used in our work corresponds approximately to $L_0 = 4/\pi \approx 1.27$. We see that increasing the lengthscale scaling factor improves the calibration

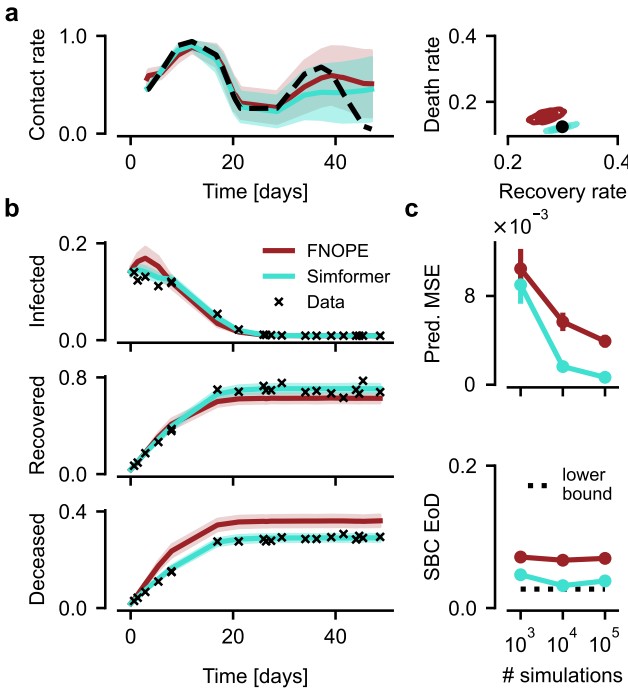

Figure S2: **SIRD model on 20 conditioning points. (a)** Posterior conditioned on 20 time points. *left:* Posterior (mean ± std.) of the time-varying parameter and ground truth parameters (dashed). *right:* Two dimensional posterior of vector-valued parameters and ground truth parameters (dot). **(b)** Posterior predictive (mean ± std.) of infected, recovered and deceased populations with observations marked. **(c)** *upper:* MSE of posterior predictive samples to observations. *lower:* Simulation-based calibration error of diagonal (SBC EoD). 'Lower bound' refers to the SBC EoD for uniformly sampled posterior ranks (details in Appendix S2). See Fig. 4 for the results with 40 conditioning points.

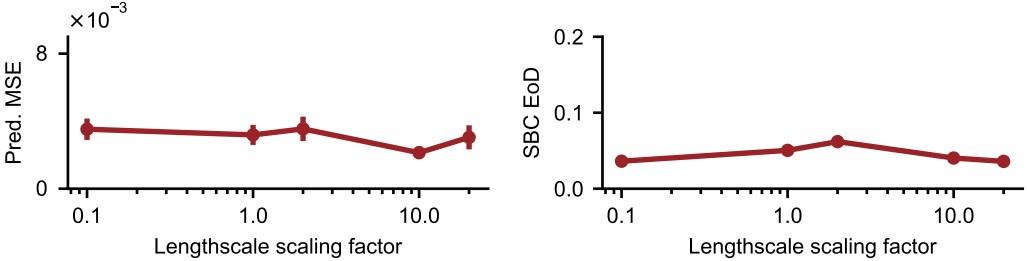

Figure S3: **SIRD ablation experiments.** Influence of the lengthscale of FNOPE's base distribution (Sec. 3.1) on posterior quality in terms of MSE of posterior predictive simulations and Simulation-based calibration error of diagonal (SBC EoD).

of the posterior learned with FNOPE. We observe that FNOPE achieves its best performance in terms of posterior log-probability when the base distribution lengthscale scaling factor is set to a value around our lengthscale heuristic (Fig. S4b).

### S7.4  Mass balance rates of Antarctic Ice Shelves

We repeat the Antarctic ice mass balance experiment (Sec. 4.5), without downsampling the parameter space from simulation to inference, i.e. inferring the full 500-dimensional posterior distribution over the mass balance parameters. Overall, we observed that FNOPE maintains its performance on this higher-dimensional problem (Fig. S5). For low simulation budgets, the performance of

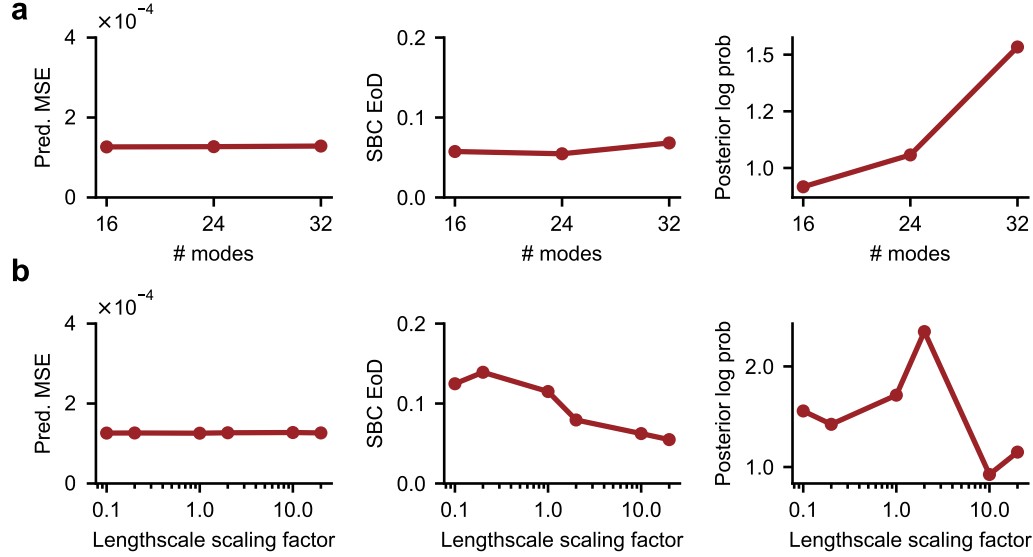

Figure S4: **Darcy ablation experiments (a)** Influence of number of used modes in the FNO block on different performance measures. The original experiments used 32 modes. Measures are the same as in Fig. 5b-d. **(b)** Influence of the noise length scale on different performance measures. The original experiments used a lengthscale of $\approx 1.27$. Measures are the same as in Fig. 5b-d.

FNOPE exceeds the other methods in terms of predictive MSE for both synthetic and real observations. As expected, spectral baseline methods significantly outperform the other baselines, especially at low simulation budgets.

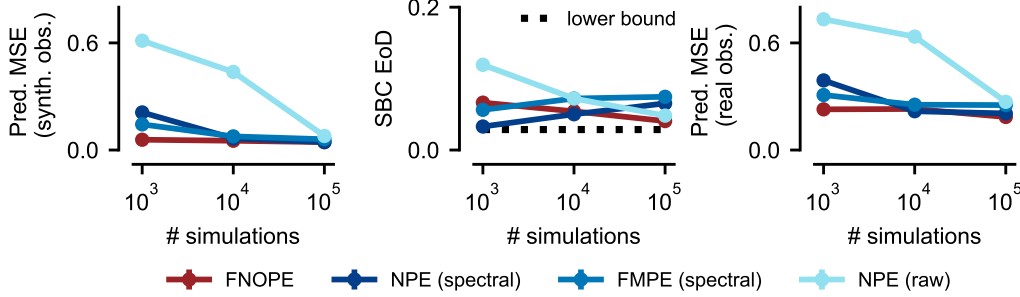

Figure S5: **Mass balance rates of ice shelves.** Inference of mass balance rates where the parameter discretization uses 500 gridpoints (observation discretization remains unchanged). Performance measures on test simulations and the real observation, where NPE (raw) refers to the method used in Moss et al. [39]. Results for FMPE (raw) omitted as this baseline was not able to always produce samples within the prior bounds across all test observations. See Fig. 6 for the results based on 50 grid points.

# S8   Additional Results

## S8.1   Additional Darcy results

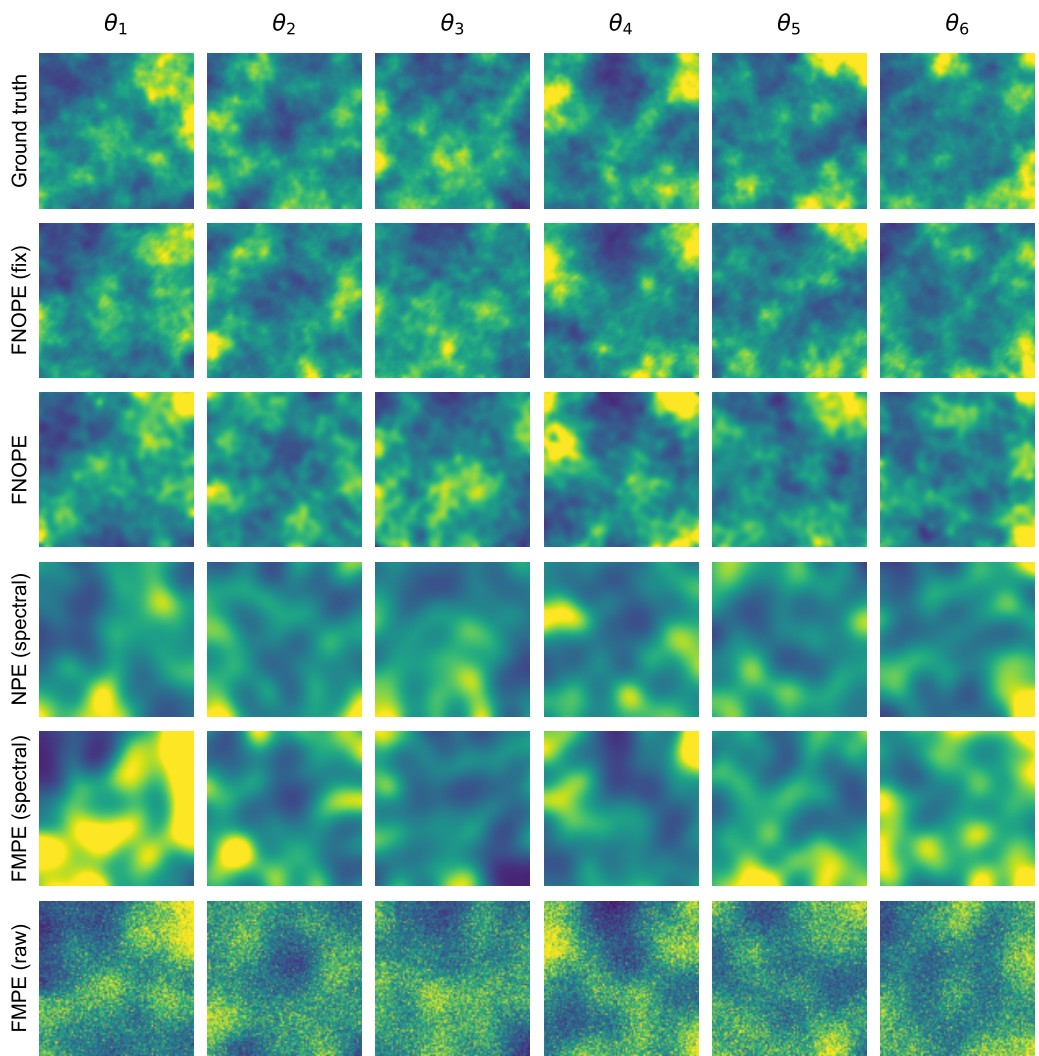

Figure S6: **Posterior samples Darcy flow experiment.** Additional posterior samples for the Darcy flow experiment for six distinct observations simulated from ground truth parameter $\theta_i$.

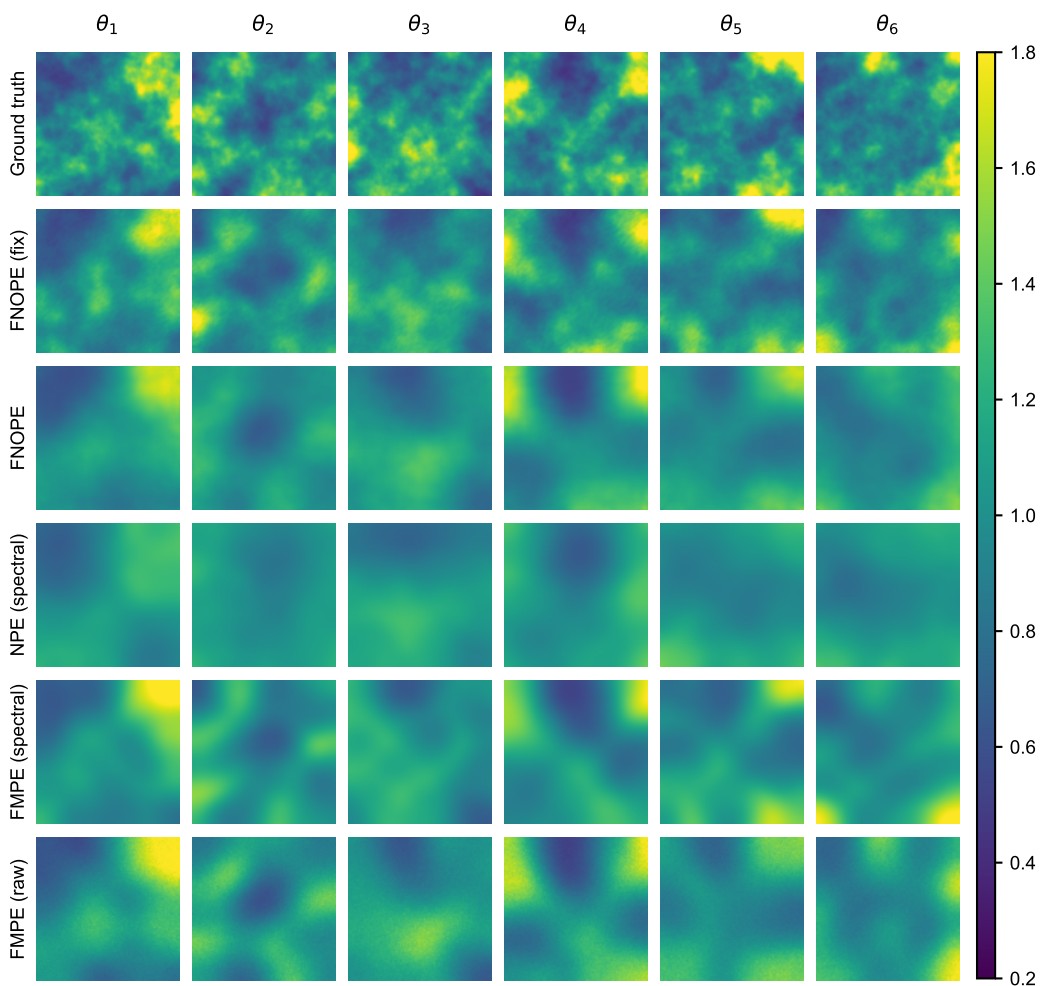

Figure S7: **Posterior means for Darcy flow experiment.** Posterior means (based on 100 samples) for the Darcy flow experiment for six distinct observations simulated from ground truth parameter $\theta_i$.

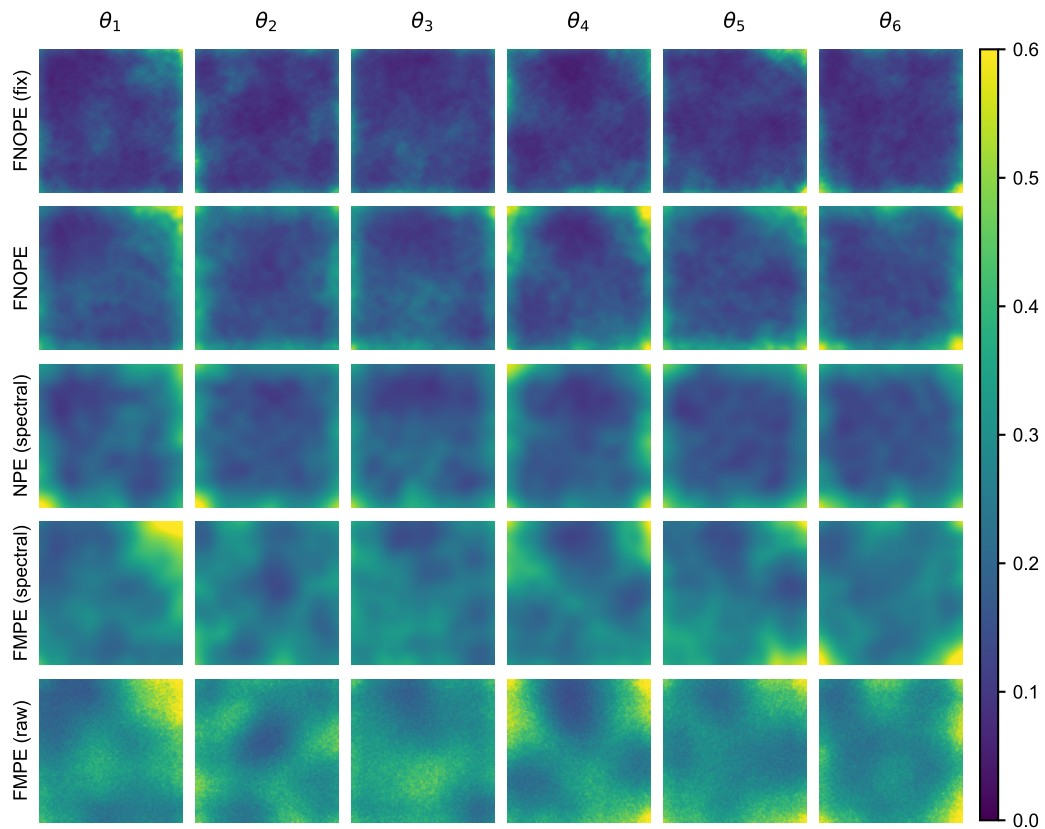

Figure S8: **Posterior standard deviation for Darcy flow experiment.** Posterior standard deviations (based on 100 samples) for the Darcy flow experiment for six distinct observations simulated from ground truth parameter $\theta_i$.

## S8.2 Comparison to invertible Fourier Neural Operator

As an additional comparison, we apply the work of Long et al. [50] and train an invertible Fourier Neural Operator (iFNO) to solve the Darcy Flow inverse problem, using the same training data and simulation budgets. We train the iFNO with the same settings as described in Long et al. [50] for the D-CURV experiment, summarized below. We observe that while iFNO achieves a comparable performance to FNOPE in terms of its predictive MSE, the posterior distribution is not calibrated as measured by the SBC EoD (Fig. S9a,b) and essentially collapsed to a point estimate. The SBC EoD value measured for iFNO (around 0.25) is consistent with this point estimate, because the one-dimensional marginal of a point mass distribution either always overestimates or underestimates the ground truth value. Hence, recalling the definition of SBC EoD (Sec. S2.2), the ground truth will have rank $r_{ij} = 1$ or $r_{ij} = K_{\text{post}} + 1$. Supposing that over different observations $x_j^o$, the point mass distribution is equally likely to underestimate or overestimate the ground truth, the cumulative distribution of the ranks will be given by $\text{CDF}_i(\alpha) = 0.5$ for all significance levels $\alpha \in (0, 1)$. Given the definition of the SBC EoD, a point mass distribution results in a SBC EoD value of

$$\int_0^1 |\text{CDF}_i(\alpha) - \alpha| d\alpha = 0.25 \tag{7}$$

for each dimension $i$. This overconfidence is also reflected in the standard deviations of the estimated distributions (Fig. S9c), which show that iFNO essentially estimates a point mass for this task.

**iFNO hyperparameters** We trained iFNO with 4 FNO blocks and 16 Fourier modes. The number of training epochs for the invertible Fourier blocks was set to 100, for the $\beta$-VAE 1000, and for joint training 100. The architecture of the $\beta$-VAE was the same as in Long et al. [50] with rank 32, as well as the value $\beta = 10^{-6}$. We used a minibatch size of 10 for joint training and 20 for the VAE and iFNO pretraining.

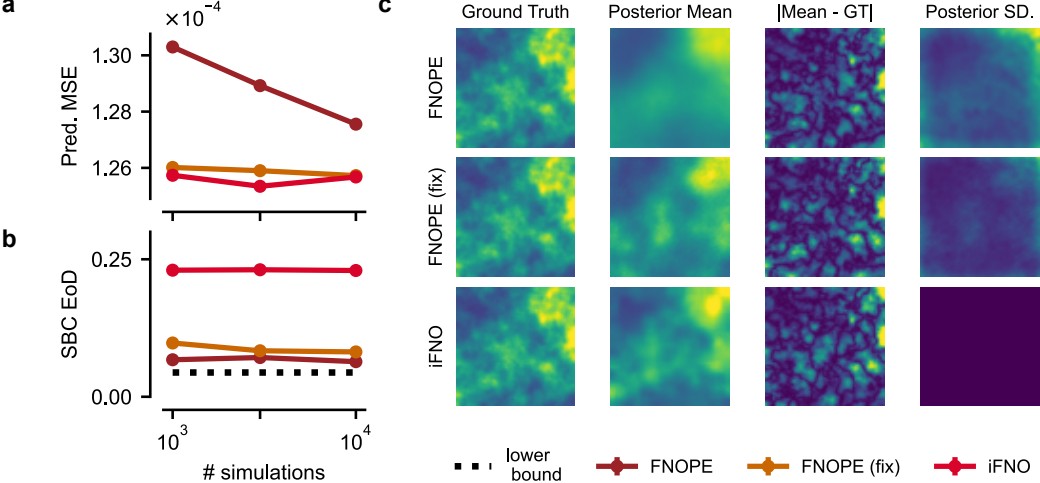

Figure S9: **Darcy Flow, comparison of FNOPE to iFNO [50]. (a)** MSE of posterior predictives to the ground truth observation (zoomed in relative to Fig. 5). **(b)** Simulation-based calibration Error of Diagonal (SBC EoD) for different training budgets. **(c)** Ground truth $\theta$, Posterior means, pixelwise error of means relative to ground truth, and posterior standard deviations. The color bars of the means match Fig. S7, and for errors and standard deviations they match Fig. S8.

