# OpenReview forum: "FNOPE: Simulation-based inference on function spaces with Fourier Neural Operators"
_NeurIPS.cc/2025/Conference — NeurIPS 2025 poster_

### Official Review · Reviewer_5iX8 · 2025-06-23

**Clarity:** 2
**Significance:** 2
**Originality:** 2
**Rating:** 3
**Confidence:** 4

**Summary:**

The paper proposes FNOPE, a simulation-based inference method that leverages Fourier Neural Operators with a flow-matching objective to infer function-valued parameters, including over irregular discretizations. The authors demonstrate the method across several benchmark and real-world scientific simulation problems. While the paper studies a problem of impacts, I find it difficult to identify a clear novelty in its core contribution.

**Questions:**

FNOPE appears to be a technically sound implementation of familiar ideas, but as presented, it lacks sufficient novelty and comparative context to justify its contributions. Please consider
- Clarify the novelty of the method in the context of existing operator-based inverse problem literature.
- Include direct comparisons with other neural operator methods.
- Conduct deeper ablation studies to isolate the impact of NUDFT, GP priors, and other design decisions.
- Strengthen the motivation for using FNOs specifically over more flexible architectures that already support irregular discretizations.

**Ethical Concerns:**

["NO or VERY MINOR ethics concerns only"]

**Final Justification:**

I’m ok with the revision and will raise my score from 2 to 3. Nevertheless, I’m surprised to hear that neural-operator approaches haven’t been applied to this specific problem yet, as the authors claim and another reviewer notes. Accordingly, I remain unconvinced about the work’s originality and novelty.

**Limitations:**

I suggest the authors to more clearly articulate the novel contributions of the paper, especially in light of extensive prior work on using neural operators for inverse problems in functional spaces. Please strengthen the positioning and clarify the methodological innovations.

**Quality:**

1

**Strengths And Weaknesses:**

Strengths: Nothing stood out as commendable.

Weaknesses:
- My main concern lies in the novelty and positioning of the contribution. Using neural operators, e.g., FNOs, to approximate mappings between function spaces is well-established. Applying them to inverse problems, including probabilistic ones, has been widely explored in recent works such as Invertible FNO (iFNO), Latent Neural Operators (LNO), and Neural Inverse Operators (NIO). If the primary novelty is the integration of NUDFT to support irregular meshes, that seems incremental, as this mechanism is neither new nor unique to this work. Several transformer-based neural operators also support irregular discretizations and are known to perform well, especially under sparse observations, as acknowledged by the authors themselves in the SIRD task (Fig. 4, pg. 6).
- Another problem is that I fail to see a fair baseline comparisons against other neural-operator-based inverse methods. It is unclear why the authors exclude these from the baseline set. Without such comparisons, it is difficult to assess whether FNOPE contributes meaningfully beyond existing neural operator architectures tailored for inverse tasks.
- While the paper includes multiple benchmarks, it lacks deeper ablations that would clarify which components are key to the method's performance. For instance, the impact of NUDFT versus FFT, the number of retained Fourier modes, or the role of GP noise shaping is not systematically studied. This makes it hard to disentangle whether FNOPE’s gains arise from architectural choices, training strategy, or task-specific tuning.

1. Long, D., Xu, Z., Yuan, Q., Yang, Y. and Zhe, S., Invertible Fourier Neural Operators for Tackling Both Forward and Inverse Problems. In The 28th International Conference on Artificial Intelligence and Statistics.
2. Molinaro, R., Yang, Y., Engquist, B. and Mishra, S., 2023, July. Neural inverse operators for solving PDE inverse problems. In Proceedings of the 40th International Conference on Machine Learning (pp. 25105-25139).
3. Kaltenbach, S., Perdikaris, P. and Koutsourelakis, P.S., 2023. Semi-supervised invertible neural operators for Bayesian inverse problems. Computational Mechanics, 72(3), pp.451-470.
4. Wang, T. and Wang, C., Latent Neural Operator for Solving Forward and Inverse PDE Problems. In The Thirty-eighth Annual Conference on Neural Information Processing Systems.
5. Behroozi, A., Shen, C. and Kifer, D., Sensitivity-Constrained Fourier Neural Operators for Forward and Inverse Problems in Parametric Differential Equations. In The Thirteenth International Conference on Learning Representations.
6. Zhongkai Hao, Zhengyi Wang, Hang Su, Chengyang Ying, Yinpeng Dong, Songming Liu, Ze Cheng, Jian Song, and Jun Zhu. GNOT: a general neural operator Transformer for operator learning. In Proceedings of the International Conference on Machine Learning (ICML), 2023.
7. Wu, H., Luo, H., Wang, H., Wang, J. and Long, M., 2024, July. Transolver: A Fast Transformer Solver for PDEs on General Geometries. In International Conference on Machine Learning (pp. 53681-53705). PMLR.

---

> ### Author Rebuttal · Authors · 2025-07-30
>
> We thank the Reviewer for the concrete suggestions. Here, we clarify the key contributions of our work, provide extensive ablation experiments and benchmark against a suggested inverse method. We hope this additional context will allow the Reviewer to rethink their score.
>
> ## Clarification of Main Contributions
> The primary contribution of our work is the development of a simulation-based inference (SBI) approach for function-valued parameters. Importantly, we clarify that Bayesian inference, in general, is _not_ equivalent to ’solving an inverse problem’: In Bayesian inference, we try to estimate the full posterior distribution over parameters, and in simulation-based inference, we aim to do this without access to likelihoods (see [1]). In contrast, inverse problems typically focus on _deterministic_ models, and aim to find a _point estimate_ that optimizes some (regularized) loss, and therefore, e.g., lack uncertainty estimates which are often crucial for scientific applications (see also response to Reviewer 5bMC). Thus, our method is simply solving a different problem to the papers pointed out by the reviewer, which either address deterministic inversion methods, such as Molinaro et al., Wang et al., and Behroozi et al. [R2, R4, R5], or on the other hand, _likelihood-based_ Bayesian inference methods, such as Kaltenbach et al. [R3].
>
> In our manuscript, we discuss related approaches that can be used for simulation-based inference of function-valued parameters, as well as how our work differs from these approaches. However, we agree with the Reviewer that clarifying the distinction between simulation-based inference and deterministic and/or likelihood-based inference in the Neural Operator context will improve the clarity of our work, and we will add this discussion in our revised manuscript.
>
> ## New baseline: Invertible Fourier Neural Operators (iFNOs)
> From the works referenced by the Reviewer, we noticed that only iFNOs [R1] can be framed as an SBI method. We thank the Reviewer for bringing up this work as including a comparison with a Neural Operator-based method will improve our paper. However, we point out that the loss presented in [R1] does not explicitly target the posterior distribution. Instead, it trains a Neural Operator-based conditional probabilistic generative model, but it is unclear whether this model learns the posterior distribution, or a different conditional distribution. To include the iFNO as a baseline in our work, we trained an iFNO on our Darcy flow problem with 1000 training simulations. While the iFNO produces reasonable samples of the parameters in terms of predictive MSE [Table R4.1], its samples lack any uncertainty. This is evident in the simulation-based calibration metric (SBC EoD, lower is better), which is close to 0.25. This value is expected for a point estimate (see discussion at the end). We also point out that this value is considerably higher than the SBC EoD values achieved by any of the methods shown in our manuscript for any task. For this experiment, we considered the low training budget regime as this is what the authors of [R1] used in their work. Furthermore, we use the same settings as in [R1] for the `D-CURV` experiment. We will include a comparison to iFNO in the manuscript.
>
> Table R4.1: Comparison of FNOPE and iFNO performance (means and standard errors) on Darcy flow task with 1000 training simulations.
> | Method ↓ / Metric →  | Predictive MSE | SBC EoD |
> |-|-|-|
> | FNOPE | 1.30e-04 ± 2e-07 | 0.067 ± 0.004 |
> | iFNO | 1.27e-04 ± 1.8e-07 | 0.232 ± 0.001 |
>
> ## Transformer-based Neural Operators
> The other two papers mentioned by the Reviewer [R6, R7] introduce new architectures for transformer Neural Operators without investigating their use for inference problems. These works are not directly comparable to our work, but using transformer Neural Operators in our inference framework as opposed to FNOs would make for an interesting approach. Our choice of the FNO architecture was motivated by the inductive bias for smoothly varying functions that FNOs leverage to obtain relatively small network sizes. In the response to Reviewer pNZ3 we provide a full breakdown of the network sizes and training times for our experiments [Table R1.1]. Even for the very large Darcy flow task ($\sim 16$k parameters), we observe that our network converged within just over an hour of training. While this network was comparably large ($\sim 11$M), we repeated the experiment in our ablations (Table R4.3), reducing the number of modes from 32 to 16. Here, we observed comparable performance despite the network size being much smaller ($\sim$3M). Nevertheless, we agree with the Reviewer that exploring transformer Neural Operators could be a beneficial future direction for our work. We will include a discussion of these papers in our manuscript.
>
> ## NUDFT vs. FFT
> We want to highlight that NUDFT is _mathematically equivalent_ to FFT if it operates on a uniform grid, whereas the FFT cannot be applied to non-uniform grids. Thus, FNOPE (fix) can be seen as an ablation of FNOPE where we replace NUDFT with FFT and add no positional noise. Additionally, we ran FNOPE on non-uniform grids, but removed the positional noise to investigate the influence of the non-uniform discretization and the corresponding processing by NUDFT. For the Linear Gaussian task there are essentially no differences of the posterior quality in terms of SWD to the ground truth posterior (Table R4.2). However, we emphasize that without positional noise, we lose the flexibility to evaluate at arbitrary positions.
>
> Table R4.2: SWD (means and standard errors) of the Linear Gaussian task with and without positional jitter.
> | Method ↓ / # simulations → | $10^2$ | $10^3$ | $10^4$ | $10^5$ |
> |-|-|-|-|-|
> | FNOPE | 0.137 ± 0.003 | 0.080 ± 0.002 | 0.033 ± 0.001 | 0.024 ± 0.000 |
> | FNOPE (fix) | 0.040 ± 0.001 | 0.028 ± 0.001 | 0.014 ± 0.000 | 0.012 ± 0.000 |
> | FNOPE (no jitter) | 0.123 ± 0.003 | 0.087 ± 0.002 | 0.035 ± 0.001 | 0.023 ± 0.000 |
>
> ## Ablation Studies
> As suggested by the Reviewer as well as Reviewers pNZ3 and u8xK, we have conducted several ablation studies to explore the effect of different hyperparameters on FNOPE. The ablations show that FNOPE performs robustly over a wide range of different hyperparameters, e.g. number of modes, lengthscale and unmasked training points. Below, we refer to tables in response to reviewer u8xK.
>
> ## Number of modes
> To investigate the influence of the number of modes, we adapted the Linear Gaussian task with a new prior, which produces samples varying at higher frequencies. Increasing the number of modes improves performance in terms of SWD as it allows the network to model higher-frequency components of the parameters more reliably (Table R3.1) and we observe a similar trend for the Darcy flow task (Table R3.2). Overall, these results highlight that the number of modes should be chosen depending on the task complexity, but the sensitivity to this hyperparameter may be limited. We refer to the response to Reviewer u8xK for a detailed discussion.
>
> ## Lengthscale of base distribution
> Varying the kernel lengthscale of the base distribution (Sec. 3.1 of paper) has a limited effect on the resulting posterior quality for the Linear Gaussian task (Table R3.3), even in the limit of a white noise base distribution.
>
> Additionally, we conducted an ablation on the Darcy flow task by varying the lengthscale of the base distribution. While the predictive MSE is robust to different lengthscales, calibration performance measured by SBC EoD is getting slightly worse with increasing lengthscale, whereas the posterior log-probability is best for intermediate lengthscales. In summary, a reasonable intermediate lengthscale scaling factor performs well across all metrics, and this is what we used in the main experiment (Table R4.3).
>
> Table R4.3: Darcy flow with 32 modes and different lengthscale scaling factors. Values reported are for 10,000 training simulations.
> | Metric ↓ / Scaling factor → | 0.1 | 0.2 | 1.0 | 2.0 | 10.0 | 20.0 |
> |-|-|-|-|-|-|-|
> | MSE | 1.26e-04 ± 1.88e-07 | 1.26e-04 ± 1.93e-07 | 1.26e-04 ± 1.83e-07 | 1.27e-04 ± 1.82e-07 | 1.27e-04 ± 1.81e-07 | 1.26e-04 ± 1.76e-07 |
> | SBC EoD | 0.125 ± 0.003 | 0.139 ± 0.003 | 0.115 ± 0.003 | 0.079 ± 0.004 | 0.062 ± 0.003 | 0.055 ± 0.003 |
> | Posterior log-prob | 1.557 ± 0.005 | 1.424 ± 0.007 | 1.714 ± 0.005 | 2.346 ± 0.007 | 0.926 ± 0.017 | 1.148 ± 0.02 |
>
> ## Number of unmasked training points
> Furthermore, we conducted an ablation study to investigate the impact of the number of unmasked training data points on the performance. In summary, FNOPE performs well for different choices of unmasked training data points. For more details on this, we refer to the response to Reviewer u8xK (Table R3.5).
>
> ## SBC EoD for a point estimate
> When training the iFNO baseline on the Darcy flow task, we obtain a Simulation-based Calibration Error of Diagonal (SBC EoD) of close to 0.25 (Table R4.1). This indicates that the distributions yielded by the iFNO are very close to point estimates. Here, we want to refer to our definition of the SBC EoD metric (Appendix S2.2). A one-dimensional marginal of a point mass distribution either always overestimates or underestimates the ground truth value. Hence, the ground truth will have rank $r_{ij}=0$ or $r_{ij}=N_\text{post}$. Supposing that over different observations $x_j^o$, the point mass distribution is equally likely to underestimate or overestimate the ground truth, the cumulative distribution of the ranks will be given by $\text{CDF}(\alpha)=0.5$ for all significance levels $\alpha\in (0,1)$. Given the definition of the SBC EoD, a point mass distribution results in a SBC EoD value of $\int_0^1 |\text{CDF}(\alpha)-\alpha|d\alpha = 0.25$.
>
> [1] Cranmer, Kyle, Johann Brehmer, and Gilles Louppe. "The frontier of simulation-based inference." Proceedings of the National Academy of Sciences (2020).

---

> > ### Comment · Reviewer_5iX8 · 2025-08-04
> >
> > I’ve reviewed the authors’ responses and appreciate their explanations. I’m willing to raise my score from 2 to 3; however, I still struggle to see the paper’s true novelty. Similar neural-network approaches to inferring function-valued parameters already exist. For example, in the Darcy case, previous networks have been validated under much more complex priors and more challenging observation settings [1,2]. With FNOPE, I find little innovation beyond swapping in an FNO. Regarding the “high-dimensional” Darcy example, I respectfully disagree: the stated dimensionality reflects the discretization, not the field’s intrinsic dimension, which should be much lower given the prior the authors define.
> >
> > 1. Padmanabha, G.A. and Zabaras, N., 2021. Solving inverse problems using conditional invertible neural networks. Journal of Computational Physics, 433, p.110194.
> > 2. Dasgupta, A., Murgoitio-Esandi, J., Ray, D. and Oberai, A., 2023, November. Conditional score-based generative models for solving physics-based inverse problems. In NeurIPS 2023 Workshop on Deep Learning and Inverse Problems.

---

> > > ### Author Response · Authors · 2025-08-05
> > >
> > > We thank the Reviewer for willing to raise their score.
> > >
> > > The two papers now referenced by the Reviewer can be framed as simulation-based inference (SBI) methods using different architectures, despite these works not linking themselves to the SBI literature. To clarify: Our claim was never to invent SBI, which we discuss and extensively compare against in our work.
> > > We therefore maintain our main contribution: We develop an efficient framework to solve SBI problems for function-valued parameter inference by combining FNO architectures with flow matching posterior estimation.
> > >
> > > As mentioned by this Reviewer, the Darcy flow problem indeed has a lower intrinsic dimensionality than the discretization in our experiments - however, the FNO architecture is precisely an architecture that can take advantage of this fact, as opposed to other architectures that do not include the same inductive bias.
> > >
> > > We thank the Reviewer again, and would be happy to answer any remaining questions.

---

### Official Review · Reviewer_u8xK · 2025-07-02

**Clarity:** 3
**Significance:** 3
**Originality:** 3
**Rating:** 4
**Confidence:** 3

**Summary:**

In this paper, the authors introduce a novel approach to Simulation-based inference (SBI) by integrating Fourier Neural Operators (FNO) with a flow matching objective. The method enables high-dimensional inference while requiring several orders of magnitude fewer simulations than traditional SBI techniques. The superior performance is demonstrated across different simulated case studies and one real-world case study.

Additionally, they are able to produce posterior samples on arbitrary discretizations and avoid reliance on any additional basis functions.

**Questions:**

1. How sensitive is the method to the number of Fourier modes retained in the FNO layers? An ablation study to assess the trade-off between accuracy and computational cost would help.
2. How dependent is the model's performance to the choice of prior distribution used? Would deviations from the assumed prior significantly degrade performance?

**Ethical Concerns:**

["NO or VERY MINOR ethics concerns only"]

**Final Justification:**

I keep my score, I recommend acceptance.

**Limitations:**

1. Several important evaluation metrics, like SWD, are not formally defined in the appendix. For example, the loss is referenced briefly on line 869 without an accompanying equation or detailed explanation.
2. Key theoretical concepts referenced in the main text (Bochner's theorem - line 120) are not formally stated or explained in the appendix.
3. The paper lacks ablation studies on several key hyperparameters. For instance, an ablation on the number of discretizations, number of fourier modes retain etc could help better understand performance bottlenecks of FNOPE.

**Quality:**

3

**Strengths And Weaknesses:**

Strengths:
1. _Innovative combination of FNO with a flow-matching training scheme:_ The FNO component effectively learns a compact representation of the data's global structure. On the other hand, the flow-matching training scheme helps perform likelihood-free inference of conditional distributions.

2. _Efficient posterior estimation in high-dimensional parameter space:_ The authors of the paper demonstrate efficient performance of FNOPE on high dimensional data under limited simulation budgets compared to FMPE, NPE and other benchmarks. Crucially, it achieves this performance without relying on fixed discretizations.

3. The authors demonstrate FNOPE's excellent performance across different synthetic benchmarks and a real-world application.

Weaknesses. Despite the strengths, the paper has certain weaknesses that need to be addressed:
1. The absence of ablation studies on key hyperparameters makes it difficult to assess the robustness of the method. Understanding this would clarify different performance bottlenecks. (Check limitations and questions for more details)

---

> ### Author Rebuttal · Authors · 2025-07-30
>
> We thank the Reviewer for their positive evaluation of our work, and for their constructive suggestions. We note that other Reviewers also acknowledge that our method can infer parameters on arbitrary discretizations (all Reviewers) and its empirical performance, especially in the low-simulation regime, (Reviewers pNZ3, 5bMC). To address the Reviewer’s main suggestion, we conducted extensive ablation studies on the hyperparameters of FNOPE. Where appropriate, the information of these tables will be expressed in figures in our revised manuscript.
>
>
> ## Investigation of hyperparameters
>
> We conducted new experiments to investigate the impact of various hyperparameters and design choices. Overall, we can show that FNOPE performs robustly over a wide range of different hyperparameters. In detail, we tested how the performance of FNOPE is affected by the number of modes of the FNO architecture, the number of unmasked training data points in training (see Sec. 3.2 of main paper), and the lengthscale of the Gaussian process used for the base distribution.
>
> ## Number of modes
>
> We investigated how the number of modes used affects the performance of FNOPE for both the Linear Gaussian and Darcy flow experiments. We augmented the Linear Gaussian task by decreasing the lengthscale of the Gaussian process prior over the parameters, resulting in a more challenging inference task with higher frequency components. The Sliced Wasserstein Distance (SWD) of the inferred posterior to the ground truth posterior improves with more modes, saturating at around 128 for both FNOPE and FNOPE (fix) (Table R3.1). This is because the FNOPE networks with fewer modes do not have the capacity to model higher-frequency components of the samples, and so cannot reproduce realistic samples. However, when increasing the number of modes, the network is sufficiently flexible to model the posterior. This is reflected in the power spectrum of the posterior samples: While samples from models with fewer modes have less power compared to the ground truth in higher frequency components, the models with 128 and 256 modes match the power spectrum of the ground truth perfectly.
>
> Table R3.1: SWD means and standard errors for FNOPE and FNOPE (fix) for the Linear Gaussian example with a simulator prior lengthscale of 0.005 and 10k training simulations.
> | Method ↓ / # modes →| 8 | 16 | 32 | 64 | 128 | 256 |
> |---|---|---|---|---|---|---|
> | FNOPE | 0.928 ± 0.006 | 0.814 ± 0.005 | 0.568 ± 0.004 | 0.251 ± 0.002 | 0.127 ± 0.001 | 0.124 ± 0.001 |
> | FNOPE (fix) | 0.933 ± 0.006 | 0.801 ± 0.006 | 0.564 ± 0.004 | 0.215 ± 0.002 | 0.035 ± 0.000 | 0.032 ± 0.000 |
>
> We observe a similar trend for the Darcy flow task, where reducing the number of modes used by FNOPE reduces the posterior log-probability the trained model assigns to data (Table R3.2). Overall, these results highlight that the number of modes should be chosen depending on the task complexity, but the sensitivity to this hyperparameter may be limited. We will provide a discussion in the main paper.
>
> Table R3.2: FNOPE performance (means and standard errors) for Darcy flow task with different numbers of modes and 10k training simulations.
> | Metric ↓ / # modes →| 16 | 24 | 32 |
> |---|---|---|---|
> | MSE | 1.26e-04 ± 1.77e-07 | 1.27e-04 ± 1.79e-07 | 1.29e-04 ± 1.85e-07 |
> | SBC EoD | 0.057 ± 0.002 | 0.055 ± 0.002 | 0.068 ± 0.003 |
> | Posterior log-prob | 0.917 ± 0.015 | 1.058 ± 0.016 | 1.534 ± 0.010 |
>
> ## Lengthscale of prior distribution
>
> We assume that when the Reviewer asks to investigate the dependence of the model to the ``prior distribution’’, they refer to the base distribution of the flow-matching formalism, as discussed in Sec 3.1 of our work. In typical inference settings, the prior over the simulator parameters is fixed and encodes any prior knowledge we have about the problem. According to Bayes’ theorem, changing the prior will also change the posterior. In contrast, the base distribution used in the flow-matching approach is a modeling choice, and can be defined freely.
>
> We investigate the impact of the choice of the base distribution on the performance of FNOPE on the Linear Gaussian and SIRD experiments. Concretely, we change the Gaussian process kernel lengthscales (Sec. 3.1), which we set proportionally to the inverse of the number of modes used for FNOPE. To vary the kernel lengthscale, we multiply it with different scaling factors. Both experiments demonstrate that FNOPE performs robustly over a range of different kernel lengthscales (Tables R3.3 and R3.4). We will provide figures illustrating this in the Appendix of the paper.
>
> Furthermore, we conducted the Linear Gaussian and the Darcy flow experiments using white noise as the base distribution instead of the Gaussian process, which would correspond to a Gaussian process with a lengthscale of zero. Both for using white noise and the Gaussian process as the base distribution, the SWD is close to zero for FNOPE and FNOPE (fix) (Table R3.3). Hence, FNOPE performs well over different base distribution choices. In the Darcy flow experiment, however, we observed that using white noise as the base distribution leads to visually noisier posterior samples. We will add figures showing this to the Appendix, but cannot include it in this rebuttal.
>
> Table R3.3: SWD means and standard errors for Linear Gaussian experiment for different kernel lengthscales and white noise as base distribution. The model used 50 modes and 10k training simulations.
> | Method ↓ / Scaling factor →   | 20 | 10 | 2 | 1 | 0.1 | white noise |
> |---|---|---|---|---|---|---|
> | FNOPE | 0.028 ± 0.000 | 0.032 ± 0.000 | 0.030 ± 0.000 | 0.031 ± 0.000 | 0.023 ± 0.000 | 0.027 ± 0.001 |
> | FNOPE (fix) | 0.019 ± 0.000 | 0.016 ± 0.000 | 0.017 ± 0.000 | 0.015 ± 0.000 | 0.015 ± 0.000 | 0.016 ± 0.000 |
>
> Table R3.4: SBC error of diagonal and predictive MSE (means and standard errors) for SIRD experiment with FNOPE for different kernel lengthscales. The model used 32 modes and 10k training simulations.
> | Metric ↓ / Scaling factor →   | 20 | 10 | 2 | 1 | 0.1|
> |---|---|---|---|---|---|
> | SBC EoD | 0.036 ± 0.003 | 0.040 ± 0.003 | 0.062 ± 0.003 | 0.051 ± 0.004 | 0.036 ± 0.003 |
> | Predictive MSE | 0.003 ± 0.001 | 0.002 ± 0.000 | 0.004 ± 0.001 | 0.003 ± 0.001 | 0.004 ± 0.001 |
>
>
> ## Number of unmasked datapoints
>
> We tested the performance of FNOPE over a wide range of unmasked training data points in the Linear Gaussian experiment. While the SWD is already small for 128 unmasked datapoints, it decreases as the number of unmasked data points increases (Tab. R3.5). With an increasing number of unmasked data points, the performance improvement starts to saturate.
>
> Table R3.5: SWD means and standard errors of the Linear Gaussian experiment for different numbers of unmasked training data points. The model used 50 modes and 10k training simulations.
> | Method ↓ / # unmasked points →   | 128 | 256 | 512 | 1000 |
> |---|---|---|---|---|
> | FNOPE | 0.090 ± 0.001 | 0.058 ± 0.001 | 0.035 ± 0.001 | 0.031 ± 0.001 |
>
> ## Additional clarifications
> ### Sliced Wasserstein Distance (SWD)
>
> We apologize for not including a formal definition of the SWD. The (empirical) sliced Wasserstein(-2) distance between $N$ samples from two probability distributions $p$ and $q$ is defined as
> $$
> \text{SWD}(p,q) = \mathbb{E}\_{u \sim U(\mathbb{S}^{n-1})} \left[ \frac{1}{N} \sum\_{i=1}^N || u^\top x\_i - u^\top y\_i ||_2^2 \right]^{1/2},
> $$
> where $x_i \sim p(x)$, $y_i \sim q(y)$ are samples from the probability distributions and $u$ are uniformly randomly sampled vectors on the unit sphere $\mathbb{S}^{D-1}$.
> We want to clarify that this metric is only used as an evaluation metric and not as a “loss” for the neural network.
>
> ### Bochner’s theorem
>
> We state Bochner’s theorem as in Williams and Rasmussen [1]:
> A complex-valued function $k$ on $\mathbb{R}^{D}$ is the covariance function of a weakly stationary mean continuous complex-valued random process on $\mathbb{R}^{D}$ if and only if it can be represented as
> $$k(\tau) = \int_{\mathbb{R}^{D}}\exp^{2\pi is\cdot \tau}d\mu(s)$$
> for some positive finite measure $\mu$.
>
> Crucially, in the less general but relevant case that $\mu$ admits a density $S(s)$, the integral is a Fourier transform between the kernel $k(\tau)$ and the spectral density $S(s)$. We apply this result to relate the lengthscale of the square exponential kernel to the spectral density of its Fourier decomposition. The definition of the SWD and the statement of Bocher’s theorem will be added to the Appendix.
>
> [1] Christopher KI Williams and Carl Edward Rasmussen. “Gaussian processes for machine learning” volume 2 (2006).

---

### Official Review · Reviewer_5bMC · 2025-07-02

**Clarity:** 4
**Significance:** 2
**Originality:** 2
**Rating:** 5
**Confidence:** 4

**Summary:**

The authors propose a method for performing simulation-based inference (SBI) of function-valued latent variables. The method is a combination of flow-matching posterior estimation (FMPE) and Fourier neural operators (FNOs): FMPE provides the objective function, while FNOs provide a means of parameterizing function-valued outputs given $x$ (plus its discretization) as input. A key innovation of the authors is their explicit conditioning on the (variable) discretization positions $l^\theta, l^x$ in both the generative (simulation) process and the inference procedure. This formulation allows for flexible choice of discretization grid, in particular the use of *any* grid desired at inference time once the model is fitted.

**Questions:**

Let me start by providing more detail on the marginalization point discussed above. Imagine a high-dimensional latent variable $\theta \in \mathbb{R}^{100}$, for example. In NPE, a common approach to avoid high-dimensional difficulties is to use a mean-field factorization, i.e. parameterize $q(\theta \mid x)$  as $q(\theta_1 \mid x) q(\theta_2 \mid x) \cdots q(\theta_{100} \mid x)$. This yields 100 univariate inference problems to solve. Certainly, fitting a hundred different networks is less than ideal, but it can be done. What is lost in this approach is posterior correlations between the latent variables, but in some settings this may be acceptable.

For the type of data in this work, losing correlations is unacceptable: if $\theta$ represents evaluations of a smooth function  on a grid, $\theta_{76}$ and $\theta_{77}$ are highly correlated, for example; they should not deviate from each other much if the grid being evaluated on is fine enough. A discussion similar to the above I think might be a useful addition to the paper, as function-valued latent variables make a compelling case to *not* use the mean-field approach with its marginalization properties. You may want to read or cite Forward Amortized Variational Inference (Ambrogioni et al.) or works that cite this one for more on marginalization.


Questions:
- The proposed method uses the FMPE objective. I wonder if you tried (or if it possible to formulate) a variant based on the NPE objective. Does this exist in other related work? Presumably it would look something like
$$
-\mathbb{E}_{l^\theta, \theta, l^x, x} \log q^\phi(\theta \mid x, l^x, l^\theta).
$$
- The section on “inferring additional parameters” seems less useful for the same marginalization argument presented above. If a user has to perform inference on both a parameter $\eta$ and a function valued latent $\theta$, I suspect in practice one might fit the mean-field variational posterior $q(\theta, \eta \mid x) = q(\theta \mid x)q(\eta \mid x)$ with the first term fit by FNOPE and the second by a standard NPE approach. Again, unless posterior correlations between $\eta, \theta$ are desired, fitting a joint posterior is unnecessary.
- The lack of discussion on marginalization clashes with the fact that SBC on the marginals is used as an evaluation metric (line 183). If the quality of the marginals is what’s being evaluated, modeling posterior correlations seems unnecessary. Adding some type of NPE competitor based on a mean-field parameterization would help resolve this point.
- Can you explain the MSE metric in more detail? What two quantities is the MSE computed between? For a fixed evaluation point $i$, is it between the ground truth draw $\theta_i$ and a single posterior sample at $i$? Or something else?
- In the first experiment, can you explain intuitively why FNOPE (fix) outperforms FNOPE?
- It’s difficult to gauge a take-away message from the second experiment given that FNOPE does not outperform the Simformer.
- In the third experiment, NPE (spectral) seems to achieve the same MSE as the proposed methods. Yet, the images in panel (a) for NPE look much poorer relative to the ground truth. Why is this?
- In the third experiment, is NPE (spectral) present in panel d), or not? If not, why?
- In the third experiment, it’s stated that FNOPE uses 32 Fourier modes, while NPE/FMPE only 16 (line 248). Should these be equal for a fair comparison?

**Ethical Concerns:**

["NO or VERY MINOR ethics concerns only"]

**Final Justification:**

See my more detailed comments below.

**Limitations:**

Yes

**Quality:**

3

**Strengths And Weaknesses:**

Strengths:

- The exposition is clear and overall the paper is well written and easily understandable.
- The use case for practitioners is an important one; NPE and other simulation-based methods are increasingly popular, and performing inference on increasingly complex quantities like function-valued latents  is a useful advancement (although I’m less familiar with other related work in the function-valued SBI space that may tackle similar problems).
- The sample efficiency point is compelling; some simulators are expensive to sample, so doing well with fewer samples is necessary.

Weaknesses:
- The paper makes claims about the difficulties of high-dimensional inference problems for SBI without talking about the marginalization properties of NPE – this is an important point that I think needs to be addressed (more on this in next section).
- The sample efficiency results are empirical in nature, rather than a theoretical guarantee. In experimental settings, it seems that NPE (spectral) can perform as well as the proposed method with enough simulations (10^4 or 10^5).
- The experimental results do not show the degree of improvement over NPE or related methods that one might expect relative to the claims. Also, some of the metrics for evaluation of the experiments raise questions about marginalization as well.
- The method is somewhat combinatorial, as both FMPE and FNOs are established approaches in the literature.

---

> ### Author Rebuttal · Authors · 2025-07-30
>
> We thank the Reviewer for their positive and thoughtful feedback. We want to highlight that the other Reviewers agree that our work solves a relevant problem (pNZ3), that the sample efficiency is a particular strength of our approach (u8xK), and noted the clarity of our presentation (pNZ3, u8xK).
>
> To address the Reviewer’s concerns, we provide a discussion on a marginalized approach as suggested by the Reviewer, which we will also add to the revised manuscript. We also provide some clarifications to answer the Reviewer’s remaining questions.
>
> ## Marginalization
> We agree with the Reviewer, that we could in principle learn a marginalized posterior in SBI. However, this is not the setting we are interested in this work. As the Reviewer mentioned correctly, a marginalized posterior is “less than ideal”, especially for function-valued parameters which are (by definition) highly correlated. But even beyond that, a lot of “low” dimensional scientific problems show highly correlated posterior distributions, and this is often what scientists care about. In neuroscience, for example, Gonçalves et al. [1] showed that the parameters of a model of the pyloric network are highly correlated, and posterior draws from the marginalized distribution fail to give reasonable simulation results. Similar properties are observed in gravitational wave detection, where correlations in the posterior are of great interest and e.g. sky position cannot be inferred with marginal inference [2]. However, we are aware that for specific problems the inference of marginal distributions might be sufficient [3]. In such cases other SBI methods might be better suited. The SIRD model pinpoints that even the low dimensional parameters $\eta$ can have strong correlations (Fig. 4a, right) and a mean field approach would suffer from these correlations. Additionally, we want to highlight that a marginalized setup would not easily allow flexible conditioning on arbitrary spatial/temporal observations as is the case for FNOPE.
>
> In terms of performance, we indeed measure the **calibration** of the inferred posteriors by SBC EoD on the marginals, which is a standard metric in the SBI literature [4]. Similar calibration measures for high dimensional distributions are still an active field of research. However, we want to highlight that all other metrics, e.g. SWD (Linear Gaussian) and posterior log-probability (Darcy flow) are taking the joint posterior into account. Similarly, the MSE metric (see clarification below) is based on posterior predictive samples from the joint posterior and is therefore indirectly also sensitive to correlation structures. Samples from a mean-field approximation often do not lead to predictive simulations which match the observation, as simulators often contain complicated, nonlinear interactions between the parameters. Consequently, our evaluation is **not** limited to the quality of the posterior marginals.
>
> We thank the Reviewer for bringing the work of Ambrogioni et al. [5] to our attention. Indeed, there are relevant related tasks for which joint posterior estimation may be circumvented via marginalized approaches. For example, if there is a large number of nuisance parameters, which are themselves not of scientific interest, estimating the joint distribution including these parameters can be very challenging. A marginalized approach as in [5] can circumvent this and learn the marginal posterior distribution over the quantities of interest, without explicitly modeling the joint distribution with the nuisance parameters. We will include this discussion on marginalization with the according references in the main paper.
>
> ## Sample efficiency
> As the Reviewer pointed out, one strength of FNOPE is that it requires fewer simulations than existing methods, which could enable inference on computationally expensive simulators. To emphasize this strength, we conducted additional experiments with 100 training simulations. See the response to Reviewer pNZ3 for more details, in particular Tables R1.2 and R1.3 show that FNOPE shows strong performance even when using only 100 training simulations in the Linear Gaussian and Antarctic Ice tasks. We agree with the Reviewer that these are empirical measurements of sample efficiency, rather than theoretical guarantees. However, this is a broad limitation in the field of SBI, and not a particular limitation of this work.
>
> ## Questions
> ### Using an NPE objective
>
> We are not aware of any NPE approach that additionally conditions on variable positions $l^x$ and $l^\theta$. Standard SBI architectures such as normalizing flows are often restricted to a fixed number of input parameters, and do not provide such flexibility. This of course stands in contrast to more recent methods such as Simformer [6], which is based on transformers. This results in a very flexible framework, but is trained with a different loss function. We also note that NPE often uses embedding nets, which can be flexibly defined without adapting the NPE loss. A Neural Operator-based embedding net could be used with a similar data-augmentation scheme as presented in our work to learn posteriors conditioned on flexibly-discretized observations. However, the discretization of the parameters would still need to be fixed (and comparatively sparse) in this scenario, as the density estimator architecture is constrained and cannot trivially be replaced by a Neural Operator.
>
> ### Joint posterior and SBC EoD on marginals
> As discussed above, we are highly interested in fitting the joint posterior, including correlation structures of the function-valued and vector-valued parameters. This is also what our evaluation metrics measure (except for calibration, where appropriate calibration measures are lacking for high dimensional spaces).
>
> ### MSE
> We apologize for the confusion regarding the MSE metric. Concretely, by MSE we refer to the MSE of the posterior **predictive**. Concretely, for each test observation $x^o_j$, we sample several posterior samples using the trained model, and for each sample, we simulate a prediction, $x_{ij}$. The MSE is then calculated between each simulation and the corresponding test observation $x^o_j$, defined as $||x_{ij}-x^o_j||^2$.  The reported MSE metric is then simply the mean of these individual MSEs over the samples $i$ from the posterior, and also averaged over several different test observations $j$. For example, for SIRD, we use 100 test observations, and 1000 posterior predictive samples for each test observation. We will make this definition clearer in our revisions. The values reported are also normalized by the number of points in the discretization of $x^o$ (full details in the Appendix S2.3).
>
> ### FNOPE vs FNOPE (fix) on Linear Gaussian
> An intuitive interpretation of the slight superiority of FNOPE (fix) in comparison to FNOPE on this task is the effect of the introduced positional noise. The noise injection modifies the likelihood, and as a result, FNOPE learns a posterior under a slightly broader likelihood than what is defined by the model. Therefore, when the true posterior is very constrained as in the Linear Gaussian task, the posterior quality is slightly poorer. However, for more challenging tasks, this is an acceptable trade off, as we gain complete flexibility on the posterior evaluation points, and the influence of the noise injection on the posterior quality is small (e.g. Fig. 5 in main paper).
>
> ### NPE (spectral) for Darcy flow
> Firstly, the Darcy flow simulator presented is a highly smoothing forward process. Even though NPE (spectral) has good performance in the predictive space (e.g. after pushing the posterior samples through the simulator), the statistics of the posterior samples might differ from the statistics of the ground truth parameters, which are drawn from the prior in this case. This is why we additionally included the posterior log-probabilities of the ground truth parameters as an evaluation metric in this task. Unfortunately, we cannot calculate the log-probabilities that the spectral methods (e.g. NPE (spectral) and FMPE (spectral)) assign to the ground truth parameters as they do not model the parameters directly. Hence, these methods are missing in Fig. 5d. We will make this more explicit in the manuscript.
>
> ### Number of modes for Darcy flow
> This is an excellent remark and we are thankful that the Reviewer brought it up. We additionally ran FNOPE with 16 and 24 modes. While there were essentially no differences for MSE and SBC EoD, the posterior log-probability slightly decreased (Tab. R2.1). We do not run the spectral baseline methods with 32 modes, as for this 2-dimensional domain this results in $32\times 32=1024$ basis functions for which these methods would need to infer the coefficients for, which is prohibitively large.
>
> Table R2.1: FNOPE performance (means and standard errors) for Darcy flow task with different numbers of modes and 10k training simulations.
> | Metric ↓ / # modes →| 16 | 24 | 32 |
> |-|-|-|-|
> | MSE | 1.26e-04 ± 1.77e-07 | 1.27e-04 ± 1.79e-07 | 1.29e-04 ± 1.85e-07 |
> | SBC EoD | 0.057 ± 0.002 | 0.055 ± 0.002 | 0.068 ± 0.003 |
> | Posterior log-prob | 0.917 ± 0.015 | 1.058 ± 0.016 | 1.534 ± 0.01 |
>
> [1] Gonçalves, Pedro J., et al. "Training deep neural density estimators to identify mechanistic models of neural dynamics." elife 9 (2020).
>
> [2] Dax, Maximilian, et al. "Real-time gravitational wave science with neural posterior estimation." Physical review letters 127.24 (2021)
>
> [3] Miller, Benjamin Kurt, et al. “Truncated Marginal Neural Ratio Estimation.” NeurIPS (2021)
>
> [4] Talts, Sean, et al. "Validating Bayesian inference algorithms with simulation-based calibration." arXiv preprint arXiv:1804.06788 (2018).
>
> [5] Ambrogioni et al., “Forward Amortized Inference for Likelihood-Free Variational Marginalization.” AISTATS (2019)
>
> [6]​​ Gloeckler, Manuel, et al. "All-in-one simulation-based inference." arXiv preprint arXiv:2404.09636 (2024).

---

> > ### Comment · Reviewer_5bMC · 2025-08-06
> > **Thanks for your detailed response.**
> >
> > Thanks to the authors; they have answered my questions and promised to address a few shortcomings I saw in the discussion. I don't have any remaining questions. I will carefully monitor the remaining discussions before coming to a final decision, but I am inclined to raise my score based on their rebuttal.

---

> > > ### Author Response · Authors · 2025-08-06
> > >
> > > We thank the Reviewer for their answer and for their positive evaluation of our work.
> > > We will incorporate all mentioned points into a revised version of our manuscript.

---

### Official Review · Reviewer_pNZ3 · 2025-07-09

**Clarity:** 4
**Significance:** 2
**Originality:** 3
**Rating:** 5
**Confidence:** 4

**Summary:**

The paper proposes to combine flow matching and Fourier neural operators to extend simulation-based inference to model posteriors over function-valued parameters, e.g., parameters that vary in time and/or space. To do this, the authors use Fourier neural operators to model the flow field and further adapt them to handle non-uniform input data using a non-uniform fast Fourier transform. The full model is able to handle both function-valued and vector-valued parameters.

The authors test the proposed model on several synthetic benchmark tasks, as well as a real-world example involving glacial flow. The method is generally on par with or outperforms baselines, requiring orders of magnitude fewer simulations to achieve the same accuracy, and having more realistic posterior samples.

**Questions:**

1. Did you measure the training and evaluation (wall-clock) times for each of the methods you consider? I'm curious to see how FNOPE compares to baselines. Also, how stable is the training?
2. The hyperparameters of the model seem to have been chosen somewhat heuristically. Did you investigate other choices? For instance, did you try other kernels for the Gaussian process prior, e.g., Matérn?
3. Further to the question above, you choose to fix the length scale of the Gaussian process kernel to a particular value, but would it not be possible to optimise it? Is there a reason not to do so?
4. Simformer seems like a strong baseline. What is the reason for not comparing against it in all of your experiments?

**Ethical Concerns:**

["NO or VERY MINOR ethics concerns only"]

**Final Justification:**

The authors addressed my questions and concerns, and provided that the submission is extended with the new experiments, I think it should be accepted.

**Limitations:**

Yes.

**Paper Formatting Concerns:**

None noticed.

**Quality:**

2

**Strengths And Weaknesses:**

**Strengths**
1. The paper presents an interesting solution to an important problem in simulation-based inference by combining two recent and promising techniques, flow matching and Fourier neural operators.
2. The model appears to work well in practice and requires many fewer simulations than the baselines, although there still are some computational concerns (see weaknesses).
3. The paper is well-written and should be of broad interest to the NeurIPS community.

**Weaknesses**
1. The main weakness of the paper is the computational complexity of the proposed model, which scales exponentially with the parameter dimensionality, both in the number of parameter discretisation points and the number of Fourier modes considered. So while the model seems like an interesting approach, it is unlikely to be useful in many scenarios in its current form.
2. The model appears to be powerful in the sense that it generally seems to require only a fraction of the number of simulations that the baselines do, but it is unclear how big the advantage is in actual compute since training and evaluation times are not reported.
3. The hyperparameters of the model seem to have been chosen heuristically and not compared to other choices.

---

> ### Author Rebuttal · Authors · 2025-07-30
>
> We thank the Reviewer for their positive and constructive feedback. We note that the other Reviewers also highlighted the applicability of FNOPE to non-uniform discretizations (all Reviewers), and were positive about the simulation-efficiency (Reviewers 5bMC, u8xK), and the clarity of our presentation (Reviewer 5bMC).
>
> To address the Reviewers’ concerns, we clarify the computational complexity which was incorrectly stated to be exponential in the number of discretization points, and report compute times, which are all in a reasonable range. Additionally, we ran extensive hyperparameter experiments to show the robustness of FNOPE across different settings.
>
> We provide tables as supporting information to our answers. Where appropriate, this information will be expressed in figures in our revised manuscript.
>
> ## Computational complexity and compute time
>
> It is not the case that the computational complexity “scales exponentially … in the number of discretization points”. Instead, in the FNO architecture with $M$ modes and $N$ discretization points, the computational complexity is only $O(N\log N)$ (if using the FFT), and $O(NM)$ (if using the NUDFT). This is one of the main advantages of the FNO architecture—since higher order frequency components are discarded, the main computational complexity of the neural network is independent of the number of points in the discretization, allowing it to be efficiently applied to coarse and fine discretizations alike. It is true that the computational complexity of the FNO scales as $O(M^D)$ in the spatial dimension $D$. However, the focus of this work is on spatial-temporal problems $D=1, 2, 3, 4$ , which cover many scientific applications, and so this exponential complexity is not a critical limitation of the FNO architecture.
>
> We believe the misunderstanding regarding computational complexity could stem from the statement in our limitations section. We apologise for this confusion, and we will clarify the distinction between the dimensionality of the spatial domains and the number of discretization points in our revisions.
>
> Additionally, we provide a full breakdown of our experiments in terms of training time, sampling time, and network sizes of FNOPE and the baseline methods (Table R1.1). Even for the largest inference task considered (Darcy Flow), training was reasonably fast at around 1 hour on a single GPU. Sampling times are larger for FNOPE on the Darcy Flow task compared to the baseline methods, mostly due to the required solution of solving high-dimensional ordinary differential equations. However, even in this case, sampling times are sufficiently fast that this difference is negligible for our applications. For all other experiments, FNOPE is faster than or equivalent to the other methods in terms of training and evaluation times.
>
> We hope that the clarification regarding computational complexities, as well as the empirical validation of the compute times, are enough to alleviate this Reviewer’s main concern regarding the applicability of our approach.
>
> Table R1.1: Network sizes and training times for the main experiments. We report the mean over 3 runs with a training budget of 10k simulations. We omit showing any variation for more readability and since the results are robust across random seeds. Linear Gaussian and SIRD tasks were run on a 2080ti GPU, Darcy flow on an A100 GPU, and Antarctic ice on CPU.
> | Task | Metric | FNOPE | FNOPE (fix) | NPE (spectral) | FMPE (raw) | FMPE (raw) | NPE (raw) | Simformer |
> | --- | --- | --- | --- | --- | --- | --- | --- | --- |
> | Linear Gaussian | # parameters | 109,797 | 108,581 | 566,590 | 299,310 | 86,302 | - | - |
> | Linear Gaussian | training (Tot.) [min] | 3.12 | 1.98 | 6.77 | 5.14 | 2.46 | - | - |
> | Linear Gaussian | training (/epoch) [s] | 0.99 | 0.87 | 2.17 | 0.31 | 0.24 | - | - |
> | Linear Gaussian | sampling time (/sample) [ms] | 2.16 | 1.06 | 0.05 | 0.20 | 0.24 | - | - |
> |  |  |  |  |  |  |  |  |  |
> | SIRD | # parameters | 116,581 | - | - | - | - | - | 285,579 |
> | SIRD | training (Tot.) [min] | 6.95 | - | - | - | - | - | 47.17 |
> | SIRD | training (/epoch) [s] | 1.19 | - | - | - | - | - | 9.43 |
> | SIRD | sampling time (/sample) [ms] | 0.22 | - | - | - | - | - | 0.22 |
> |  |  |  |  |  |  |  |  |  |
> | Darcy flow | # parameters | 11,555,673 | 11,552,521 | 3,537,566 | 9,180,033 | 897,664 | - | - |
> | Darcy flow | training (Tot.) [min] | 72.24 | 41.83 | 20.52 | 12.20 | 10.95 | - | - |
> | Darcy flow | training (/epoch) [s] | 20.30 | 26.23 | 2.82 | 0.73 | 0.66 | - | - |
> | Darcy flow | sampling time (/sample) [ms] | 279.55 | 35.20 | 0.21 | 2.70 | 1.95 | - | - |
> |  |  |  |  |  |  |  |  |  |
> | Antarctic ice | # parameters | 25,317 | - | 235,608 | 96,210 | 92,340 | 360,933 | - |
> | Antarctic ice | training (Tot.) [min] | 47.35 | - | 19.69 | 33.53 | 31.63 | 6.80 | - |
> | Antarctic ice | training (/epoch) [s] | 9.82 | - | 4.21 | 2.02 | 1.92 | 5.64 | - |
> | Antarctic ice | sampling time (/sample) [ms] | 22.79 | - | 1.50 | 23.85 | 16.61 | 1.53 | - |
>
>
> ## Simulation efficiency
>
> As the Reviewer pointed out, one strength of FNOPE is that it requires fewer simulations than existing methods, which could enable inference on computational expensive simulators. To emphasise this strength, we conducted experiments using only 100 training simulations. For the Linear Gaussian task the SWD is considerably smaller for FNOPE and FNOPE (fix) than for the baseline methods with 100 simulations (Table R1.2). Similarly, on the Antarctic ice task, FNOPE is more robust than the baseline methods (Table R1.3), even though the performance degrades for all methods, .
>
> Table R1.2: SWD means and standard errors of the Linear Gaussian experiment for different numbers of training simulations.
> | Method ↓ / # simulations →   | $10^2$              | $10^3$              | $10^4$              | $10^5$              |
> |----------------------------------|-------------------|-------------------|-------------------|-------------------|
> | FNOPE                            | 0.118 ± 0.002   | 0.144 ± 0.002   | 0.089 ± 0.001 | 0.086 ± 0.000 |
> | FNOPE (fix)                      | 0.042 ± 0.001 | 0.026 ± 0.000 | 0.016 ± 0.000 | 0.010 ± 0.000 |
> | NPE (spectral)                     | 1.237 ± 0.008   | 1.084 ± 0.006   | 0.683 ± 0.008  | 0.034 ± 0.001 |
> | FMPE (raw)                         | 1.053 ± 0.005   | 0.944 ± 0.002  | 0.884 ± 0.000 | 0.874 ± 0.000 |
> | FMPE (spectral)                    | 1.306 ± 0.008   | 1.102 ± 0.008   | 0.821 ± 0.008   | 0.172 ± 0.008 |
>
> Table R1.3: Predictive MSE means and standard errors of the Antarctic ice experiment for different numbers of training simulations.
> | Method ↓ / # simulations → | $10^2$ | $10^3$ | $10^4$ | $10^5$ |
> |---|---|---|---|---|
> | FNOPE | 0.108 ± 0.002 | 0.069 ± 0.002 | 0.054 ± 0.001 | 0.046 ± 0.001 |
> | NPE (spectral) | 0.355 ± 0.006 | 0.180 ± 0.003 | 0.071 ± 0.002 | 0.062 ± 0.001 |
> | FMPE (raw) | 0.244 ± 0.003 | 0.238 ± 0.003 | 0.074 ± 0.001 | 0.058 ± 0.002 |
> | NPE (raw) | 0.246 ± 0.003 | 0.287 ± 0.004 | 0.093 ± 0.002 | 0.045 ± 0.001 |
> | FMPE (spectral) | 0.365 ± 0.005 | 0.165 ± 0.004 | 0.091 ± 0.002 | 0.075 ± 0.002 |
>
>
>
> ## Extensive Hyperparameter Ablations
>
> Overall, FNOPE performs robustly over a wide range of different hyperparameters (see response to Reviewer u8xK for more details and result tables). First, we performed an ablation study on the impact of the Gaussian process kernel lengthscale on the performance for the Linear Gaussian and SIRD experiments (Tables R3.3, R3.4), which shows that FNOPE performs well for different lengthscales, even in the limit of white noise. Second, we investigated the dependence on the number of Fourier modes used for the Linear Gaussian and the Darcy flow experiment (Table R3.1, R3.2): FNOPE’s performance increases with the number of modes, but saturates for a higher number of modes. Third, we investigated the effect of the number of unmasked training data points on the performance of our method (Table R3.5). Here, we focus on the Linear Gaussian experiment. FNOPE performs well for all numbers of unmasked training points, but the performance increases slightly with the number of unmasked training points and exhibits saturating behavior with higher numbers of unmasked points. Figures for these ablation studies will be included in the Appendix of the paper.
>
>
> ## Different base distributions (e.g. Matérn kernel)
>
> Usually, the base distribution in flow-matching models is the unit Gaussian, $\mathcal{N}(0,I)$. Our main motivation in introducing the Gaussian Process base distribution is to produce base distribution samples which satisfy the smoothness assumptions of our FNO architecture, while still providing a distribution which can be efficiently sampled and evaluated. Thus, the main design choice in the base distribution is the power spectrum of the samples, which can be controlled via the lengthscale (see discussion above on hyperparameter ablations). A Matérn kernel could also present a valid choice of kernel, but we do not expect it to lead to different results.
>
> ## Simformer as baseline
> The computational cost of Simformer is at least $O(N^2)$ in the number of parameters, which is infeasible for a very large number of points as in the Darcy Flow Task ($\sim 16$k).
>
> Our goal in comparing FNOPE to Simformer on the SIRD task is to demonstrate that FNOPE is comparable to an established approach at the challenging task of inferring parameters on non-uniform positions, as well as inferring additional vector-valued parameters. More specifically, for the SIRD experiment, we infer the parameters on much sparser grids than the rest of our experiments (40 gridpoints, compared to e.g. 1000 for the Linear Gaussian). However, FNOPE can be scaled to finer discretizations in contrast to Simformer.

---

> > ### Comment · Reviewer_pNZ3 · 2025-08-05
> >
> > Dear authors,
> >
> > Thank you very much for your rebuttal, which addressed my questions and concerns. I am impressed by your efforts to extend the ablation studies and compile the new tables -- they provide a much clearer picture for me. I am satisfied that FNOPE is sufficiently original and practical, and I will therefore raise my score.

---

> > > ### Author Response · Authors · 2025-08-06
> > >
> > > We thank the Reviewer for their constructive review and for raising their score. We are happy that we could provide a clearer picture of our work.

---

### Note · Authors · 2025-08-12

In this work, we introduce FNOPE, an approach for performing simulation-based inference of function-valued parameters by using Fourier Neural Operators (FNOs) trained with a flow matching objective. We show on several benchmark tasks, as well as one real-world task, that our approach enables SBI for function- and vector-valued parameters simultaneously, and can be flexibly applied to non-uniform discretizations.

Several Reviewers identified the performance of FNOPE for low simulation budgets as an advantage of our work (pNZ3, 5bMC, u8xK), and noted the relevance of the problem FNOPE addresses (pNZ3, 5bMC) and the clarity of our work (pNZ3, 5bMC, u8xK) as strengths.

During the rebuttal period, we performed extensive ablation studies as requested by several reviewers to improve our work. We varied the number of frequency modes used by FNOPE, the lengthscale of the Gaussian Process base distribution in the flow matching scheme, and the masking and positional noise used during training, in order to investigate performance bottlenecks and sensitivity to these hyperparameters. We found that the performance of FNOPE is robust to all of these hyperparameters within a reasonable range. Only when the number of modes is very low is the FNO block unable to capture high-frequency components, which could lead to performance degradation.

Additionally, we clarified the difference between our approach and other FNO-based approaches for inverse modeling (in response to Reviewer 5iX8). In summary, the main distinction is that our approach performs simulation-based inference, meaning we infer the full posterior distribution over the parameters, without requiring access to likelihood evaluations from the simulator. We additionally ran a comparison to one of the works mentioned by Reviewer 5iX8, and found that while it produces good parameter estimates in terms of their predictive simulations, the uncertainties are not well calibrated. We will include this comparison in our work.

Overall, we could address all major concerns during the rebuttal phase. The Reviewers were convinced by our additional ablation and FNO experiments, with three of the four Reviewers explicitly indicating a willingness to increase their score. We will include all of the material presented during the rebuttal phase in an updated version of our manuscript.

We thank the Reviewers again for their constructive reviews and engagement with our work.

---

### Decision · Program_Chairs · 2025-09-17

**Decision:**

Accept (poster)

**Comment:**

Reviewers all find merit in this paper, praising the problem as important (Reviewer PNZ3, Reviewer 5bMC), the writing as clear and high-quality (Reviewer 5bMC), and the combination of Fourier neural operators and simulation-based inference as interesting (Reviewer pNZ3, Reviewer u8xK). Most of the weaknesses I saw had to do with minor details about experimental evaluation, and the fact that this is a paper whose main contribution is to combine two existing methods, making it potentially incremental (Reviewer 5iX8). I am not convinced that incrementalness is a problem here, because learning in function spaces, compared to learning over finite-dimensional vectors, is sufficiently-rich that extensions to this domain are by their nature not that straightforward. The other downside mentioned is lack of theoretical guarantees (Reviewer 5bMC), but this weakness applies to the entire research areas this paper covers, rather than just this paper - and these areas have proven themselves useful in practice, so this criticism is similarly not too convincing.

On basis of reviewer consensus, I recommend acceptance of this work